# Semantic F1 Scores: Fair Evaluation Under Fuzzy Class Boundaries

## Abstract

We propose **Semantic F1 Scores**, novel evaluation metrics for subjective or fuzzy multi-label classification that quantify semantic relatedness between predicted and gold labels. Unlike the conventional F1 metrics that treat semantically related predictions as complete failures, Semantic F1 incorporates a label similarity matrix to compute soft precision-like and recall-like scores, from which the Semantic F1 scores are derived. Unlike existing similarity-based metrics, our novel two-step precision-recall formulation enables the comparison of label sets of arbitrary sizes without discarding labels or forcing matches between dissimilar labels. By granting partial credit for semantically related but nonidentical labels, Semantic F1 better reflects the realities of domains marked by human disagreement or fuzzy category boundaries. In this way, it provides fairer evaluations: it recognizes that categories overlap, that annotators disagree, and that downstream decisions based on similar predictions lead to similar outcomes. Through theoretical justification and extensive empirical validation on synthetic and real data, we show that Semantic F1 demonstrates greater interpretability and ecological validity. Because it requires only a domain-appropriate similarity matrix, which is robust to misspecification, and not a rigid ontology, it is applicable across tasks and modalities.

## 1 Introduction

Multi-label classification in subjective or conceptually rich domains, like emotion recognition or identifying expressions of moral foundations, frequently involves labels that are semantically interrelated or even interchangeable in some settings. Standard evaluation metrics such as F1 scores (Fujino et al., 2008; Loza Mencía et al., 2023) treat any inexact match as a complete failure, even when the predicted label is close to the "gold" label (e.g., *anger* and *disgust*). In practice, researchers and practitioners routinely apply hard evaluation metrics to subjective classification tasks because they represent the current standard and are compatible with existing evaluation pipelines (Alhuzali & Ananiadou, 2021; Chochlakis et al., 2023a; Sabour et al., 2024; Lian et al., 2025). Crucially, in these domains there is often no single "correct" answer or objective "ground truth". It is typically substituted with "crowd" truth (Aroyo & Welty, 2015; Resnick et al., 2021), relying on the wisdom of the crowd. Disagreement is the norm, and errors are also common (Chochlakis et al., 2025).

To address these issues, we introduce **Semantic F1**, a family of metrics that extends the standard F1 measure by granting partial credit proportional to the semantic similarity between predicted and gold labels in multi-label settings, which yields a label similarity matrix. Given such a label similarity matrix, we then compute a two-step match: (i) map each prediction to its closest gold label (semantic precision), (ii) map each gold label to its closest prediction (semantic recall). We next combine them via harmonic mean, mirroring classic F1, to derive **sample, micro, and macro Semantic F1 scores**. This two-step process is also shown in Figure 1 (and in 8 for a hierarchical label tree). Our novel two-step design avoids common pitfalls of previous single-step algorithms (Kuhn, 1955; Sun & Lim, 2001; Turki et al., 2020) for semantic evaluation metrics, as it accounts for both **over-prediction** (semantically unrelated predictions) and **under-coverage** (missing label coverage) of the semantic label space. Importantly, Semantic F1 is grounded in existing evaluation theory. In the special case where no partial credit is desired, Semantic F1 scores reduce exactly to the conventional F1 scores.

Beyond the theoretical framing, we validate Semantic F1 across eight synthetic and real-data studies. We show that hard F1 fails to separate provably worse predictors, whereas Semantic F1 decays lin-

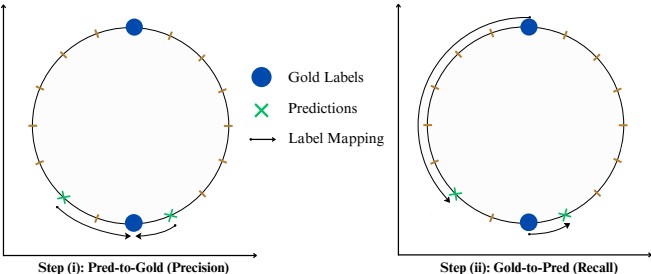

Figure 1: Qualitative demonstration of the matching method used between two sets of labels. Each prediction is mapped to its closest gold label, and vice versa, capturing **over-prediction** and **under-coverage** respectively. For visualization purposes, labels are placed on a unit circle, in a metric space where distance is measured based on angles, similar to (Plutchik, 1980)'s wheel of emotions.

early with both error rate and magnitude, remains robust to partial misspecification of the similarity matrix, and operates successfully in non-metric spaces by capturing cross-manifold errors. Notably, it avoids failure modes of existing semantic metrics. We also test LLMs on subjective tasks including predicting emotions (metric), moral foundations (non-metric), and downstream negotiation outcomes. Semantic F1 better reflects model performance, correlates more strongly with downstream outcomes, and behaves more intuitively under varying thresholds. Finally, using Semantic F1 for early stopping consistently yields superior generalization across both hard and semantic metrics.

Semantic F1 is built for practical deployment. It supports the common case of discrete label predictions and requires only a similarity matrix, which can be precomputed from domain knowledge, correlations, or embeddings. Unlike regression-based similarity measures, it naturally compares sets of *arbitrary* size, making it directly applicable to multi-label regression as well.

Our contributions can be summarized as follows: we introduce a novel suite of metrics, Semantic F1 scores, that leverage semantic relatedness to evaluate predictors. We show that our 2-step formulation is interpretable and has intuitive theoretical properties and no failure modes, compared to previous work. We also mathematically show that Semantic F1 scores fallback to the conventional F1 scores when no partial credit is assigned, providing extra guarantees of robustness. We conduct extensive experiments to validate the utility of the proposed metrics compared to existing work.

## 2 RELATED WORK

### 2.1 ONTOLOGY-DRIVEN SEMANTIC EVALUATION

Turki et al. (2020) proposed constructing confusion matrices that align predicted and gold labels using ontology-driven similarity, thereby enabling semantically weighted evaluation in multi-label settings. While this approach awards partial credit, it relies on a single alignment direction and conditions on the relative size of the two sets. As a result, it fails to penalize both over-prediction and under-coverage symmetrically, producing biased or incomplete assessments. Bansal et al. (2022) proposed a similar algorithm, but aimed at single-label sentence retrieval. An alternative is the Hungarian algorithm (Kuhn, 1955), which enforces one-to-one matches between predictions and gold labels. This constraint often discards legitimate predictions or gold labels, leading to unintuitive penalties, and its natural extensions introduce additional failure modes. In contrast, our two-step formulation maps directly onto precision and recall, preserving the robustness guarantees of the F1 score, avoiding the above pitfalls. Thus, it subsumes Turki et al. (2020) and resolves the limitations of Hungarian-style and other single-step similarity-based algorithms, as discussed in §3.2. Hierarchical multi-label classification approaches use semantics, where matches can be extended to larger neighborhoods in a hierarchical tree (Sun & Lim, 2001; Amigo & Delgado, 2022).

### 2.2 COST-SENSITIVE MULTI-LABEL LEARNING

Bénédict et al. (2021) introduced sigmoidF1, a smooth and differentiable approximation of F1 that can be optimized during training. Similarly, Lin (2023) introduced probability-guided losses. Both

maintain exact matches between labels. Rossi et al. (2018) introduced SML, leveraging label similarity during model training. Alhuzali & Ananiadou (2021); Chochlakis et al. (2023a); Huang et al. (2024) used label relationships at the example level during training. He & Xia (2018); Chochlakis et al. (2023a) used the cosine similarity of emotional representations for regularization. Hierarchical multi-label learning has also used graph-based similarity (Ramírez-Corona et al., 2016; Amigo & Delgado, 2022). Semantic F1 score, in contrast to all these approaches, introduces an algorithm to holistically match sets of labels in multi-label settings, as opposed to the exact matches, even if soft.

### 2.3 LABEL SIMILARITY

Efforts to model label similarity span theoretical, embedding-based, and metric learning approaches. In emotion research, labels are often embedded in psychological spaces such as Plutchik (1980)'s wheel of emotions or the Circumplex model by Russell (1980), sometimes extended with Dominance or higher-dimensional representations (Demszky et al., 2020). These spaces provide direct measures of similarity between emotions and are commonly applied in regression-based recognition for single-label settings, where the label space is assumed metric. By contrast, moral foundations exemplify *multiple, separable clusters*: classic Moral Foundations Theory distinguishes *binding* (Loyalty, Authority, Purity) from *individualizing* (Care, Fairness) foundations (Graham et al., 2009; 2011), and recent work further splits Fairness into *Equality* and *Proportionality* (Atari et al., 2023).

Embedding methods have also been deployed. Word2vec (Mikolov et al., 2013) and GloVe (Pennington et al., 2014) embeddings enable similarity-based classification (Alhuzali & Ananiadou, 2021; Chochlakis et al., 2023a), even in zero-shot scenarios (Wang et al., 2018; Chochlakis et al., 2023b). More recently, metric-learning approaches have explicitly optimized similarity structure. LIMIC (Mao et al., 2023) learns both global and label-specific metrics, while LSMM (Mao et al., 2024) extends this to multiple local metrics informed by semantic or clustering partitions. Hierarchical approaches instead exploit the label graph itself, using structural relationships to define similarity (Sun & Lim, 2001; McFee et al., 2017; Falis et al., 2021; Amigo & Delgado, 2022).

Despite this progress, no prior work has developed a principled semantic match for the predicted and gold label sets while preserving the interpretability and robustness of the F1 score. Semantic F1 fills this gap by integrating semantic structure without sacrificing theoretical soundness.

## 3 SEMANTIC F1 SCORE

### 3.1 PROBLEM FORMULATION

Consider a multi-label classification problem where we have input samples $\mathbf{x} \sim P(\mathcal{X})$ from an input space $\mathcal{X}$, and each sample is associated with a set of labels $\mathbf{y} \subseteq \mathcal{L}$, where $\mathcal{L} = \{\ell_1, \ell_2, \ldots, \ell_m\}$ is the universe of possible labels. $\mathbf{y}$ is a set because multi-label settings can include none, one, or multiple labels in no particular order. Given a dataset $\mathcal{D} = \{(\mathbf{x}_i, T_i)\}_{i=1}^n$ where $T_i \subseteq \mathcal{L}$ represents the ground truth label set for example $i$, a classifier predicts label sets $P_i \subseteq \mathcal{L}$ for each input $\mathbf{x}_i$. In many real-world scenarios, the ground truth labels $T_i$ may be noisy, incomplete, or come from unreliable annotators. Moreover, traditional evaluation metrics (e.g., conventional F1) treat all label mismatches equally, failing to account for semantic relationships between labels. To address this, we introduce semantic relatedness through a semantic similarity matrix $\mathbf{S} \in [0, 1]^{|\mathcal{L}| \times |\mathcal{L}|}$, where $S_{ab}$ quantifies the semantic similarity between labels $\ell_a$ and $\ell_b$. This similarity can be derived from label co-occurrence statistics, embedding distances, domain-specific knowledge graphs, or by using theoretical work in the domain (see §B).

### 3.2 CORE ALGORITHMS

**Matching Function** Let $A, B \subseteq \mathcal{L}$. The fundamental building block is the best matching function that computes the average semantic similarity of label set $A$ to $B$:

$$\text{BestMatch}(A, B, S) = \frac{1}{|A|} \sum_{a \in A} \max_{b \in B} S_{a,b}. \tag{1}$$

This function, asymmetric in $A$ and $B$, finds the best semantic match in set $B$ for each element in set $A$, then returns the arithmetic mean of these maximum similarities. For all $a \in A$, we denote their

best match in $B$ as $M_{A,B}(a)$. Intuitively, this matching function awards partial credit for semantic proximity in the space defined by $S$ rather than treating all mismatches as zero, identifying how well the space occupied by $A$ is semantically covered by $B$. Its time complexity is $O(|\mathcal{L}|^2)$, a small cost for most scenarios. We expand in §3.6 on how this runtime can be reduced, if required.

**Pointwise Semantic F1 Score**    For a single example with predicted set $P_i$ and gold set $T_i$, we apply the matching function in both directions, from $P_i$ to $T_i$, and from $T_i$ to $P_i$. Matching predictions to gold labels corresponds to what we define as **Semantic Precision**, as it quantifies how close the predictions are to some positive class in the semantic label space and, therefore, ignoring false negatives. As a result, it penalizes **over-prediction** in the label space, the semantic equivalent of false positives. On the other hand, matching gold labels to predictions corresponds to a **Semantic Recall**, quantifying how well gold label semantics are captured by some prediction in label space, while ignoring false positives. This is the semantic equivalent of capturing false negatives, which we refer to as **under-coverage**. Pointwise Semantic Precision, Recall, and F1 are defined as:

$$\text{Precision}_i^s = \text{BestMatch}(P_i, T_i, S), \tag{2}$$

$$\text{Recall}_i^s = \text{BestMatch}(T_i, P_i, S), \tag{3}$$

$$\text{SeF1}_i = H(\text{Precision}_i^s, \text{Recall}_i^s) = \frac{2 \cdot \text{Precision}_i^s \cdot \text{Recall}_i^s}{\text{Precision}_i^s + \text{Recall}_i^s}. \tag{4}$$

This is necessary to ensure that both **over-prediction** and **under-coverage** are penalized. Figure 1 also qualitatively demonstrates why both directions are essential in multi-label settings. In Figure 1(i), one gold label is unmatched; ignoring it would ignore under-coverage. Conversely, by flipping predictions and gold labels in Figure 1, the recall step only would leave one prediction unmatched, ignoring over-prediction. Alternatively, we could match all labels within a single step. A single step, however, does not have the interpretability of separate precision and recall steps, and it has to *force* unrelated matches, which intuitively is an undesirable property. We further elaborate on this and other theoretical limitations of existing single-step similarity-based approaches in §A.

## 3.3 Sample Semantic F1 Score

The sample Semantic F1 score is defined as the arithmetic mean of the pointwise scores:

$$\text{SeF1}_{\text{samples}} = \frac{1}{n} \sum_{i=1}^{n} \text{SeF1}_i. \tag{5}$$

A key edge case occurs when the similarity matrix is equal to the identity matrix $I_{|\mathcal{L}|}$ (denoted simply as $I$ henceforth), and no partial credit is given to inexact predictions (this can happen, for example, when every label pair is distant in label space; we further discuss the interpretation of $S = I$ in §C). Then, we can easily derive that:

$$\text{BestMatch}(A, B, I) = \frac{|A \cap B|}{|A|} \Rightarrow \text{SeF1}_i = \frac{2 \cdot |P_i \cap T_i|}{|P_i| + |T_i|}, \tag{6}$$

making $\text{SeF1}_{\text{samples}}$ equivalent to the sample F1 score (Fujino et al., 2008; Loza Mencía et al., 2023) when $S = I$. Thus, the conventional (hard) sample F1 is a special case of our formulation. Likewise, without partial credit, semantic precision and recall also reduce to their conventional definitions.

## 3.4 Micro Semantic F1

The micro approach aggregates semantic relatedness across all examples before computing the F1 score. We define the semantic true positives, false positives, and false negatives as:

$$\text{TP}_i = \sum_{p \in P_i} S_{M_{P,T}(p),p} \qquad \Rightarrow \text{TP} = \sum_{i=1}^{n} \text{TP}_i \tag{7}$$

$$\text{FP}_i = \sum_{p \in P_i} (1 - S_{M_{P,T}(p),p}) \qquad \Rightarrow \text{FP} = \sum_{i=1}^{n} \text{FP}_i \tag{8}$$

$$\text{FN}_i = \sum_{t \in T_i} (1 - S_{t,M_{T,P}(t)}) \qquad \Rightarrow \text{FN} = \sum_{i=1}^{n} \text{FN}_i \tag{9}$$

Micro-averaged precision and recall are then computed in the usual way, yielding:

$$\text{SeF1}_{\text{micro}} = \frac{2 \cdot \text{TP}}{2 \cdot \text{TP} + \text{FP} + \text{FN}}. \tag{10}$$

As with the sample-based formulation, when $S = I$, the micro Semantic F1 reduces exactly to the conventional (hard) micro F1, and so do the micro Semantic Precision and Recall.

### 3.5 MACRO SEMANTIC F1

For macro-averaging, we compute per-class Semantic F1 scores and average them. For each class $c \in \mathcal{L}$, we accumulate semantic counts across all examples in which it appears:

$$\text{TP}_c = \sum_{i=1}^{n} S_{M_{P,T}(c),c}, \ \text{FP}_c = \sum_{i=1}^{n} 1 - S_{M_{P,T}(c),c}, \ \text{FN}_c = \sum_{i=1}^{n} 1 - S_{c,M_{T,P}(c)} \tag{11}$$

The per-class and macro Semantic F1 scores are then defined as:

$$\text{SeF1}_c = \frac{2 \cdot \text{TP}_c}{2 \cdot \text{TP}_c + \text{FP}_c + \text{FN}_c} \Rightarrow \text{SeF1}_{\text{macro}} = \frac{1}{|\mathcal{L}|} \sum_{c \in \mathcal{L}} \text{SeF1}_c. \tag{12}$$

Again, setting $S = I$ collapses the macro Semantic F1 to the standard (hard) macro F1 score. We cover edge cases and other variants in §D, with pseudocode in §G.

### 3.6 EXTENSION TO CONTINUOUS SPACES

So far, we have assumed fixed discrete gold and predicted labels, which permits the $O(|\mathcal{L}|^2)$ runtime and explicit precomputation of the similarity matrix. A key novelty of our approach is that it extends seamlessly to continuous semantic spaces without fixed label sets. This generalization not only broadens applicability but can also reduce runtime in the discrete case. Specifically, assuming $a, b \in \mathbb{R}^d$, the matching can be reformulated as

$$\text{BestMatch}(A, B, D) = \frac{1}{|A|} \sum_{a \in A} \max_{b \in N(a,B)} \hat{S}(a, b; D), \tag{13}$$

where $D$ is a distance measure (e.g., Euclidean for a metric space, Isomap (Tenenbaum et al., 2000) for a non-metric space) used to compute similarity $\hat{S}$ online, such as $\hat{S}(a, b; \|.\|_2) = \frac{1}{1+\|a-b\|_2}$. $N$ retrieves nearest neighbors of $a$ in $B$, typically faster than brute-force $O(|B| \cdot d)$ search (Wang et al., 2024). This extension enables principled and interpretable evaluation in multidimensional multi-label regression and even classification, for instance when using prototypical embeddings (Papaioannou et al., 2025). Semantic F1 can then be computed as usual.

## 4 EXPERIMENTS

### 4.1 DATASETS

**SemEval 2018 Task 1 E-c (Mohammad et al., 2018)**  Subjective multi-label emotion recognition of 11 emotions. We use the English tweets. We refer to this as **SemEval** for short. Similarity is derived as normalized cosine similarity from Plutchik (1980)'s metric wheel.

**GoEmotions (Demszky et al., 2020)**  Subjective multi-label emotion recognition benchmark of 27 emotions. Similarity is derived from train set correlations, which might not generalize to other settings due to the distinction between *semantic* and *associative* relations, which we discuss in §B.

**MFRC (Trager et al., 2022)**  Subjective multi-label moral foundation corpus from Reddit for six moral foundations. The majority of examples contains no labels, so similarity is derived from correlations in Atari et al. (2023). This dataset exemplifies a clustered, non-metric label space: classic Moral Foundations Theory distinguishes binding from individualizing foundations (Graham et al., 2009; 2011), with recent work further splitting them (Atari et al., 2023). As a result, MFRC provides a natural testbed for evaluating Semantic F1 in non-metric, multi-manifold domains where partial credit is meaningful within clusters but less so across them.

**PersuasionForGood (Wang et al., 2019)**   Negotiation dialogues in which the persuader attempts to convince their interlocutor to donate part of their task earnings to charity. Donation amounts for persuader and persuadee are provided. We formulate a binary prediction task, *persuader success*, denoting whether the persuadee donates more than the persuader, evaluated using the ROC-AUC score. Because it does not include emotion annotations, we manually annotated two examples per taxonomy (GoEmotions or SemEval) for use in prompting. Emotions are then predicted per turn, given previous turns as context. We adopt an 85–15 train/test split. We refer to this as **P4G**.

## 4.2 IMPLEMENTATION DETAILS

We use the 4-bit quantized versions of the open-source LLMs through the *vLLM* (Kwon et al., 2023), *HuggingFace* (Wolf et al., 2020) and `bitandbytes` interface for *Py-Torch*. We use GPT-4.1 (`gpt-4.1-mini`), GPT-4o (`gpt-4o-mini`), Llama-2 7B and 70B (`meta-llama/Llama-2-#b-chat-hf`), and Llama-3 8B and 70B (`meta-llama/Llama-3.#-#B-Instruct`). We set set temperature to 0. We use random retrieval of examples. We finetune BERT (`bert-base-uncased`; Devlin et al. 2018) using Demux (Chochlakis et al., 2023a). Standard splits are used if not specified. More details in §E.

## 4.3 RESULTS

To substantiate the utility of our proposed metric, we present four synthetic-data and four real-data studies. They address two key questions: (i) do the theoretical advantages of Semantic F1 translate to actual improvements in controlled, synthetic settings over hard F1 and baseline semantic-based metrics? and, (ii) does it provide more meaningful evaluation than current practice on real-world tasks? We use the synthetic settings to precisely control the behavior (the amount of error) of the predictions, which allows us to make precise claims about the behavior of the metrics across provably worse predictors. The real-world tasks verify our theoretical findings and our results from the synthetic experiments in practice, using real data to supplement our synthetic results. The intuition behind real-world tasks is that a more informative metric will correlate better with other interesting aspects of the model and the data, providing indeed a better automated evaluation of models.

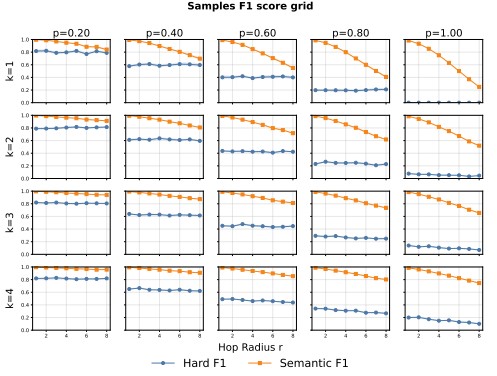

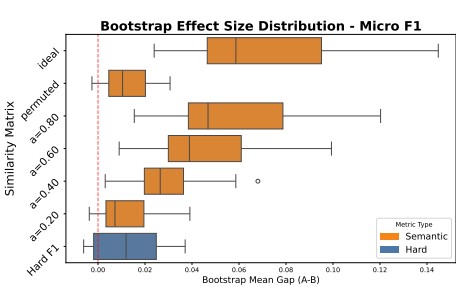

(a) Hard vs semantic sample F1 score across number of labels $k$, perturbation probability $p$, and hop radii $r$.

(b) Distribution of micro F1 differences between Near and Far predictions, aggregated across perturbation probabilities, label number, and radii.

Figure 2: Semantic metrics scale with worse predictors, even with moderately misspecified similarity matrix, in contrast to hard metrics.

### 4.3.1 SYNTHETIC STUDY A: SYNTHETIC CONSTRUCT VALIDITY

**Motivation.**   We test whether Semantic F1 properly reflects the semantic closeness between gold labels and predictions in a controlled setting, namely labels placed on a ring in a 2D space (similar to Plutchik (1980)'s wheel, see the individual components in Figure 14). We also evaluate the robustness to different degrees of misspecification of the similarity matrix. To do so, we add random noise in the non-diagonal elements of the similarity matrix and check whether the semantic metrics still sufficiently differentiate between provably better and worse classifiers. This would indicate that

knowing only roughly the similarity between labels is sufficient for Semantic F1 to serve as a good evaluation metric.

**Setup.** We arrange $n = 24$ labels on a unit circle at angles $\theta_i = 2\pi i/n$ and define the ground-truth similarity $S$ via normalized cosine similarity $S(i, j) = 0.5 + \cos(\theta_i - \theta_j)/2 \in [0, 1]$. Gold sets are generated by sampling $k$ labels per example from a cosine-peaked distribution around a random center. Concretely, we first sample one label uniformly at random, then sample the remaining $k-1$ labels with probabilities proportional to their similarity to the first label. We construct two families of perturbed predictors: *Near-miss* and *Far-miss*. For each gold label and perturbation probability $p$, we substitute the label with one exactly $r$ hops away on the ring: $r \in \{r_{\text{near}}\}$ for Near, $r \in \{r_{\text{far}}\} > r_{\text{near}}$ for Far. We sweep $k \in \{1, 2, 3, 4\}$, $p \in \{0, 0.2, 0.4, 0.6, 0.8, 1\}$, and hop radii $r_{\text{near}} \in \{1, 2, 3, 4\}$, $r_{\text{far}} \in \{5, 6, 7, 8\}$. To assess robustness to similarity matrix misspecification, we evaluate under: (i) ideal $S$, (ii) a row-permuted, invalid $S$, and (iii) mixtures $S_\alpha = \alpha S + (1 - \alpha)U$ with $\alpha \in \{0.8, 0.6, 0.4, 0.2\}$, where $U$ is Gaussian noise of $0.5$ deviation.

**Metrics and statistics.** We report hard F1 using `sklearn` and Semantic F1 (micro/macro/samples). We show how these metrics vary across hop radii $r$ and perturbation probabilities $p$ with 1000 examples per configuration. For sensitivity to true difficulty, we compute Kendall's $\tau$ between metric value and the true radius $r$ (expect decreasing with $r$). For Near vs. Far comparisons, we bootstrap the mean gap (A−B) with $B = 25$ resamples for 95% CIs.

**Results.** Figure 2a shows sample F1 as a function of the number of gold labels $k$, perturbation probability $p$, and hop radius $r$, with mean and 95% CIs from bootstrapping. Hard metrics remain largely invariant to hop radius and may even increase as predictions become less semantically relevant, indicating sensitivity to noise rather than semantic closeness. By contrast, Semantic F1 decreases monotonically with both hop radius and perturbation probability. Corresponding Kendall's $\tau$ statistics are reported in Figure 11 of §F.1. For Near vs. Far comparisons, Figure 2b (micro F1) shows the distribution of metric differences between $r_{\text{near}}$ and $r_{\text{far}}$. Semantic F1 provides much clearer separation than hard F1. Moreover, Semantic F1 is robust to moderate to high misspecification: even mixtures with $\alpha = 0.2$ maintain separation comparable to that of the hard metric. However, when the similarity matrix is fully permuted, separation collapses, eliminating any advantage over hard F1. Other metrics and full experimental grids are presented in §F.1.

### 4.3.2 SYNTHETIC STUDIES B-D

We present results for synthetic studies B through D in the appendix, §F.2, §F.3, and §F.4 respectively. Briefly, synthetic study B examines heuristic bimodal classifiers, showing that hard metrics favor such simple heuristics over near-miss classifiers, unlike Semantic F1. Synthetic study C investigates non-metric spaces: hard F1 remains invariant to within- and across-manifold jumps, even when the latter should be penalized, whereas Semantic F1 captures this distinction, even under moderate misspecification. However, when similarity is naively constructed under metric assumptions, Semantic F1 also becomes insensitive to cross-manifold errors, highlighting the need for caution in non-metric settings. Finally, synthetic study D demonstrates beyond the theoretical arguments in §3.2 and §A that other similarity-based baselines fail to separate good from bad predictors. The failure cases of semantic precision and recall are similar to the ones of their conventional counterparts, justifying the switch to the Semantic F1 score as a generic evaluation metric in domains without special considerations. For this reason, we focus on the Semantic F1 score for the following real-data studies. It is important to note, however, the precision and recall might be more appropriate for domains where under-coverage or over-prediction respectively are not important. The extended Hungarian algorithm conflates over-prediction and under-coverage, creating failure modes that Semantic F1 does not have, making its usage in generic or special domains hard to justify.

### 4.3.3 REAL STUDY A: MULTI-LABEL THRESHOLD BEHAVIOR

**Motivation.** We test whether Semantic F1 varies more smoothly with decision thresholds and yields more stable model rankings than standard hard F1 when applied to supervised multilabel heads. This would indicate that Semantic F1 is more robust to threshold choice, making evaluations less sensitive to hyperparameter choice.

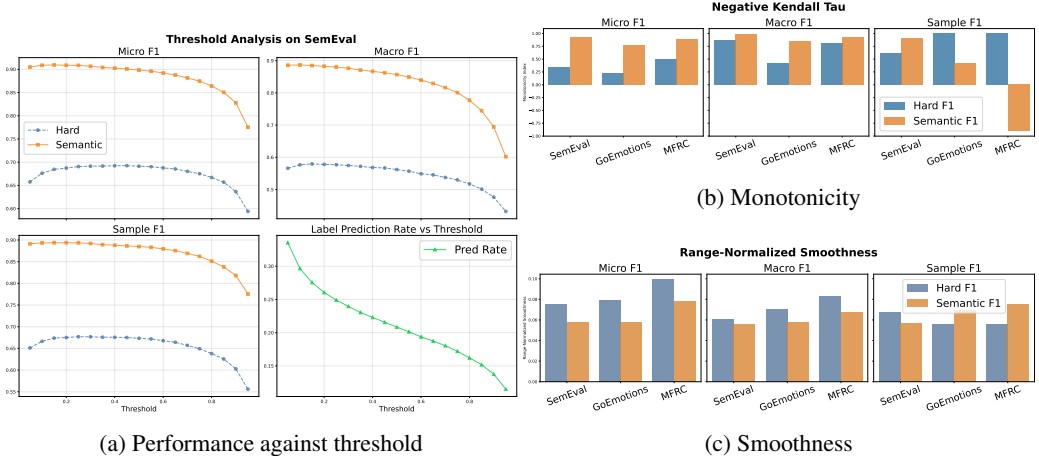

(a) Performance against threshold

(b) Monotonicity

(c) Smoothness

Figure 3: Threshold analysis on Demux

**Setup.** We evaluate multi-label classifiers, namely Demux and Llama-3 1B with trained classification head, on subjective multi-label datasets. For each dataset we fix a similarity matrix $S$, and for each model we sweep decision thresholds $\tau \in \{0.1, 0.2, \ldots, 0.9\}$, binarize predictions and compute metrics at each $\tau$.

**Metrics and statistics.** At every threshold we report hard F1 and Semantic F1 (micro/macro/samples). We summarize smoothness and ranking behavior across the threshold grid $\mathcal{T} = (\tau_1, \ldots, \tau_T)$ with two indices: (i) Monotonicity: Kendall's $\tau$, reported as its negative value (so that more decreasing monotonic trends correspond to higher values, which we expect; thus, higher is better) (ii) Smoothness: the average absolute step change between consecutive thresholds, normalized by the value range (lower is better, since large jumps indicate "bumpiness"). Using absolute differences disentangles smoothness from monotonicity, while range normalization accounts for pragmatic scale differences between metrics. By convention, a higher monotonicity index and lower smoothness index indicate greater robustness to threshold choice.

**Results.** Figure 3 shows the threshold behavior of Demux on SemEval (with Llama-3 1B and full dataset results in §F.5). Semantic F1 declines more monotonically as thresholds increase, reflecting that lower probabilities often indicate semantically related but inexact labels, information captured by Semantic F1 but ignored by hard F1. Quantitative comparisons in Figure 3b and 3c show that across datasets and models, Semantic F1 consistently achieves a higher monotonicity index and lower smoothness index, demonstrating greater robustness to threshold variation and more stable model rankings than hard F1.

### 4.3.4 REAL STUDY B: ECOLOGICAL VALIDITY

**Motivation.** We test whether semantic similarity in subjective predictions matters for downstream applications. Specifically, we ask whether models ranked higher by Semantic F1 on subjective tasks produce emotional features that better predict outcomes in an objective downstream task than models ranked by conventional hard F1. Better correlation with downstream tasks signals a more informative and ecologically valid metric.

**Setup.** Using the SemEval and GoEmotions taxonomies, we predict emotions at every turn of negotiation dialogues with six LLMs, conditioned on prior turns. The predicted emotions are then used as features for logistic regression models that predict negotiation outcomes. Features are constructed by averaging predictions over the last $k$ turns, and we report the best downstream performance across $k \in \{1, 2, \ldots, 10\}$. Performance on the downstream task is compared to each model's performance on the source emotion dataset (from which the taxonomy is drawn), using 2-shot prompts for both.

**Metrics and statistics.**   We repeat each logistic or linear regression experiment with 100 different seeds and LLM inference 5 times for a subset of 300 test examples each to derive means and 95% confidence intervals. We report Spearman correlation between downstream performance and source-task performance across models.

**Results.**   Figure 4 shows that Semantic F1 correlates more strongly with downstream performance than hard F1. In particular, Figure 4a demonstrates that downstream outcomes are almost perfectly correlated with Semantic F1 (with $p < 0.01$), whereas correlation with hard F1 is absent. More broadly, across all settings (see §F.6), Semantic F1 is at least as predictive of downstream performance as hard F1. These findings highlight the ecological validity of Semantic F1: semantically similar, though not identical, emotion predictions yield quantitatively similar downstream effects, making Semantic F1 a better proxy for real-world utility. Notably, when using either only semantic precision or only semantic recall, that is not the case, as shown in §F.6.

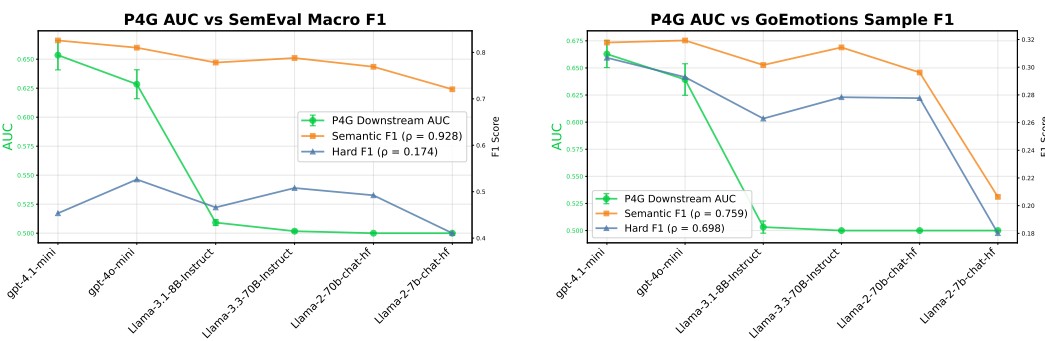

(a) Correlation between macro F1s and downstream with SemEval

(b) Correlation between sample F1s and downstream with GoEmotions

Figure 4: Ecological validity study results comparing semantic and hard F1 correlation with downstream performance across different emotion datasets. X axis ordered by downstream performance.

### 4.3.5   REAL STUDY C: EARLY STOPPING CRITERION

**Motivation.**   We test whether Semantic F1 as early stopping criterion leads to better generalization than using hard F1 scores. The hypothesis is that predictors optimized for semantic similarity will generalize better, even if ultimately evaluated on hard matches.

**Setup.**   We train and evaluate Demux on SemEval, GoEmotions, and MFRC. We use either a hard F1 or a Semantic F1 metric as an early stopping criterion on the development set, and then evaluate the best model based on these on the test set.

**Metrics and statistics.**   We evaluate each model on seven metrics: Semantic and hard F1 (micro/macro/samples; six in total) plus the Jaccard score. Each experiment is repeated 10 times, and we report means with 95% CIs. For comparisons between semantic-based and hard-based early stopping, we compute two-sided p-values testing equality of the resulting distributions.

**Results.**   Table 1 shows the number of metrics (development and test) on which runs using Semantic F1 for early stopping outperform those using hard F1, and vice versa. Semantic-based stopping consistently yields more favorable outcomes, both in terms of trends and statistically significant gains, compared to hard-based stopping. Particularly on MFRC, we see that with hard early stopping, only the criterion metric itself shows a significant improvement, while with semantic early stopping, gains extend even to Jaccard score, a purely hard metric. Full results across datasets are presented in §F.7. Overall, these findings indicate that semantic metrics provide superior early stopping criteria, producing models that generalize better across both semantic and hard metrics.

Table 1: Early stopping method comparison: Wins across all metrics (6 F1s and Jaccard Score). Statistically significant wins are shown in parentheses. Ealry stopping criterion was sample F1.

| Dataset | Dev | | Test | |
|---|---|---|---|---|
| | Semantic | Hard | Semantic | Hard |
| GoEmotions | 7 (0) | 0 (0) | 2 (0) | 5 (0) |
| MFRC | 2 (2) | 5 (0) | 4 (2) | 3 (1) |
| SemEval | 4 (0) | 3 (0) | 7 (0) | 0 (0) |
| **Total** | **13 (2)** | **8 (0)** | **13 (2)** | **8 (1)** |

### 4.3.6 REAL STUDY D: CONVERGENT VALIDITY

We hypothesize that Semantic F1, when applied to subjective tasks, better reflects the performance improvements of newer LLMs on objective tasks than hard F1. As shown in §F.8, Semantic F1 indeed tracks objective performance more faithfully: while Spearman correlations are comparable, its Concordance Correlation Coefficient (CCC) is substantially higher, indicating not only alignment in trends but also a much closer match in absolute values across a variety of similarity matrices.

## 5 CONCLUSION

We introduce the *Semantic F1* scores, a principled extension of the multi-label F1 score that incorporates semantic similarity between labels while retaining the interpretability and robustness of precision–recall reasoning. Our two-step formulation resolves the shortcomings of prior single-step and Hungarian-style approaches by penalizing both over-prediction and under-coverage without forcing spurious matches or discarding labels. Crucially, when no partial credit is assigned, Semantic F1 collapses exactly to standard F1, ensuring backward compatibility with existing pipelines, and extending the standard F1's robustness to similarity-based metrics.

Through numerous studies, we demonstrate the advantages of Semantic F1. In controlled experiments, it scales smoothly with semantic error, distinguishes near- from far-miss predictors, and remains robust under moderate misspecification of the similarities, including in non-metric spaces. In real-world evaluations of LLMs, Semantic F1 produces more stable threshold behavior, stronger alignment with downstream outcomes, and improves generalization when used for early stopping. These findings establish Semantic F1 as both theoretically sound and practically effective.

One limitation of Semantic F1 lies in its dependence on the similarity matrix, as for any semantic metric. Although our experiments (§4.3, §F) show robustness to moderate misspecification and diverse initializations, we recognize that poorly designed similarity matrices (e.g., dense, adversarial, or culturally inappropriate) may degrade interpretability or fairness, as we also show in §F.3. In practice, we recommend constructing similarity matrices from out-of-sample correlations, validated embeddings, or domain ontologies derived from work of domain experts, and inspecting their sparsity and scaling before deployment, following §B and insights from §F.3 and §F.4. Future work could explore methods to further mitigate sensitivity. Importantly, when similarity reduces to the identity matrix, Semantic F1 gracefully collapses to standard F1, ensuring a safe fallback. Additionally, the framework should treat similarity matrices as culturally and contextually variable, rather than as universal structures (Atari et al., 2023).

Taken together, our synthetic and real-data results show that Semantic F1 provides a fairer and more informative evaluation for subjective and fuzzy classification tasks if appropriate semantic similarities between labels are used, offering a drop-in replacement for hard F1 that better reflects ecological validity and downstream utility. We believe that this metric fills a critical gap in evaluation methodology and can serve as a foundation for future work on semantically and culturally grounded subjective performance measures. We expect that each community will converge to a single or a fixed of similarity matrices to use for semantic evaluation, and urge dataset distributors to also release similarity matrices for their datasets as the canonical similarity to be used during evaluation, or clearly cite the theoretical work motivating their taxonomy.

## ETHICS STATEMENT

Even when semantic similarity is properly captured, practitioners should *not* treat similarity judgments as universal and should acknowledge their cultural and contextual variability. Different populations demonstrate systematically different intuitions about semantic similarity: Atari et al. (2023), where we derive our similarity matrix for morality, demonstrate so for moral foundations across cultures, for example. Constant or sole reliance on established psychological models (e.g., Plutchik (1980)'s wheel) or training correlations can hardcode one cultural perspective as universal, creating systematic bias toward the population whose judgments the similarity matrix reflects. This limitation is particularly problematic for AI systems deployed across diverse populations, where evaluation fairness requires acknowledging that semantic relationships themselves are culturally constructed rather than objectively given. Used properly, Semantic F1 can actually act as another measurement tool for cultural bias, measuring preference for ontologies that emerge in specific cultures rather than others.

Nevertheless, to address these limitations during evaluation, a methodological framework is needed that treats similarity matrices as empirically validated, population-specific instruments rather than fixed universal structures. This approach requires three components: (i) collecting human similarity judgments from the target population through psychometric studies, e.g., where participants rate label pairs on standardized scales, (ii) learning calibration functions that map embedding distances to these human-derived similarity scores, and (iii) generating population-specific similarity matrices that reflect the actual conceptual relationships meaningful to the intended user community. Once validated, these calibration functions could be deployed beyond the original label set, in related tasks for that same population.

## REPRODUCIBILITY STATEMENT

We provide several resources to ensure clarity and reproducibility of our methods. Pseudocode for all core algorithms is included in §G to avoid any ambiguity in implementation. Each study contains dedicated **Setup** and **Metrics and statistics** sections, detailing experimental design and evaluation to facilitate reproduction. In §B, we give explicit instructions for constructing similarity matrices, and in §E we provide prompts and additional implementation details, complementing §4.2 in the main text. Finally, we commit to releasing our code, including Semantic F1, baseline methods such as the extended Hungarian, and all experimental pipelines, upon publication.

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

## A    SINGLE-STEP MATCH

To illustrate the limitations of relying on only one direction of alignment (precision or recall), we present worst-case scenarios in Figure 5. Algorithms that condition on cardinality, such as Turki et al. (2020), are not immune: arbitrarily many predictions may cluster around a single gold label, or vice versa, particularly in continuous spaces. Even the examples in Figure 1 (and its flipped version) already expose the weaknesses of single-step precision or recall, without resorting to extreme cases.

Beyond single-step precision or recall, the Hungarian algorithm (Kuhn, 1955) is another potential approach. However, because it enforces one-to-one matching between equinumerous sets, it must discard one of two equally close but inexact predictions by assigning it to a dummy zero match. This produces unfair penalties, even when all relevant subspaces of the gold label space are covered without over-prediction (Figure 6). Similar issues when using Optimal Transport or Wasserstein distance (Montesuma et al., 2024) to measure the similarity between two sets with arbitrary sizes due to the constraint of deriving a single distribution from the set.

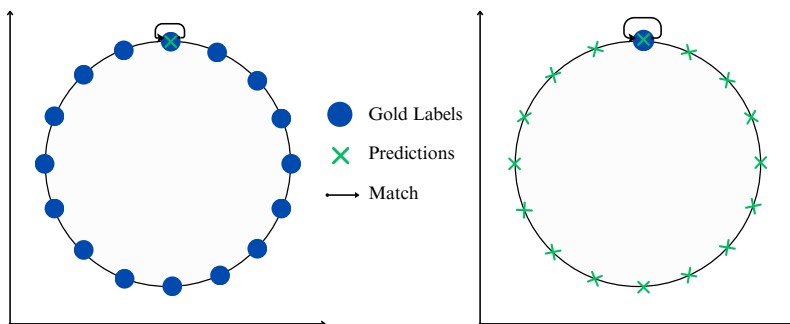

Figure 5: Worst-case scenarios for using only precision (left) and recall (right)

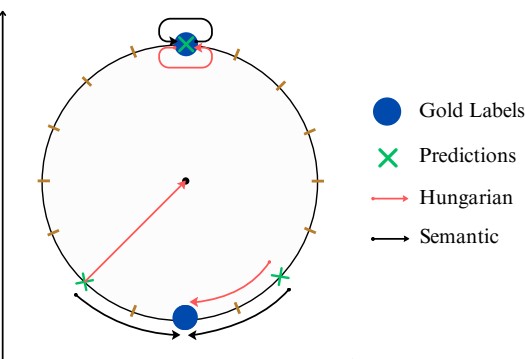

Figure 6: Comparison between our matching algorithm and the Hungarian match: The Hungarian algorithm requires equinumerous sets, thus discarding one of the two equidistant predictions of the model, an unfair penalty.

A natural extension is to patch the Hungarian algorithm by assigning unmatched labels to their closest neighbor. Yet this hybrid approach sacrifices interpretability and introduces new failure modes. Under an arithmetic mean, unmatched gold labels can be drowned out by artificially many correct matches, allowing the metric to be gamed through over-prediction (Figure 7i). Using a harmonic mean avoids this but over-penalizes a single missed label (Figure 7ii). Extending Optimal Transport to unbalanced scenarios (Montesuma et al., 2024) where sets might have different cardinalities requires balancing many hyperparameters on top of the similarity matrix.

Our Semantic F1 avoids these pitfalls. By separating over-prediction and under-coverage into two interpretable steps (semantic precision and semantic recall) and combining them with a harmonic mean, it faithfully balances errors on both sides. When the similarity matrix is the identity, this formulation collapses exactly to standard F1, preserving its robustness while extending it into the semantic domain. Unlike single-step or Hungarian-style approaches, our method does not assume equal cardinality of label sets, nor does it force spurious matches to distant labels. Every prediction and every gold label contributes to the final value, enabling a more nuanced and fine-grained evaluation.

In summary, previous similarity-based metrics suffer from unfair penalties, unintuitive averaging, or restrictive assumptions. Our two-step Semantic F1 retains the interpretability and grounding of precision and recall, extends the robustness of F1 into semantic settings, and provides a more faithful evaluation of multi-label predictions.

## B  How to Build your Similarity Matrix

In this section, we go into more detail about the ways we created the similarity matrix in this work.

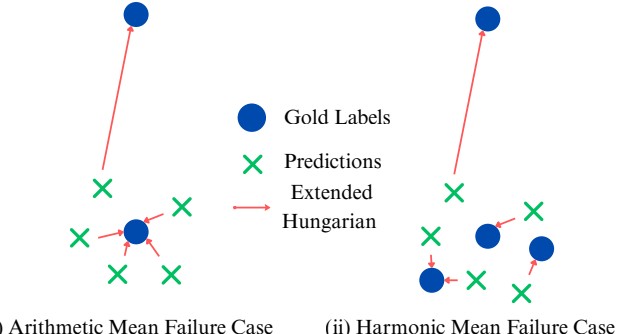

(i) Arithmetic Mean Failure Case     (ii) Harmonic Mean Failure Case

Figure 7: Example failure modes of potential extensions for the Hungarian algorithm. (i) Using the arithmetic mean drowns out under-coverage; (ii) The harmonic mean overly penalizes for a single missed gold label. Note that the clustering of the gold labels in case (ii) is done for visual purposes only, and not required for it to be a failure mode. In contrast, our two-step algorithm (i) uses recall to isolate under-coverage from the correct predictions, and (ii) averages out the missed gold label with the many covered gold labels in the recall step.

**Euclidean distance.** We can use $\|.\|_2$ to compute the distance between embeddings in what was assumed a metric space (§F.3). There, we used standard practice to convert the Euclidean distance to a similarity score as $S(x, y; \|.\|_2) = 1/(1+\|x-y\|_2)$. Since $\|.\|_2 \geq 0 \Rightarrow S(x, y; \|.\|_2) \in [0, 1]$, with $S(x, y; \|.\|_2) \to 0$ when $\|x - y\|_2 \to +\infty$. Based on the baseline differences between embeddings, a scaling factor may be appropriate to amplify or dampen the distance: $S(x, y; \|.\|_2) = 1/(1 + \beta\|x - y\|_2)$ in order to, in turn, amplify or dampen the difference in partial credit between labels. For instance, in a space where $\min_{(a,b)\in\mathcal{L}^2} \|a - b\|_2 = 10$, $\beta$ can be set to $1/10$ to make partial credit stronger in the space, as for practical purposes even a minimum distance of 10 might result in $S \approx I$ for all practical intents and purposes. Moreover, the embeddings can be normalized, or lower and higher-order distances can be used, like $\|.\|_1, \|.\|_3, \ldots, \|.\|_\infty$, that might produce more meaningful similarity values. This method allows us to use, e.g., word embeddings for the labels in a setting, and construct a similarity matrix. It is also suitable for use in regression settings, as it can be computed online for each example.

**Cosine similarity.** Cosine similarity is used similarly to measure similarity (distance) in a metric embedding space. Its range of values is $[-1, 1]$, hence we simply normalize to $[0, 1]$ as $0.5 + s/2$. Again, scaling can be applied to the values depending on how quickly we want the values to go to 0, for example by squaring or cubing the normalized values. A constant normalization factor $1/\beta$ results in perfectly aligned embeddings (meaning even identical embeddings) having a similarity of $1/\beta$, which is usually not desirable.

**Correlations** The correlations should be computed outside the evaluation set to ensure the generalization of the evaluation. Similar to cosine similarity, we use an affine normalization to map the values to the $[0, 1]$ range. One interesting side-effect of using correlations is that this is a purely data-driven method, whereas the previous two may be applied to a theoretical space like Plutchik (1980)'s wheel of emotions. Moreover, it does not assume a metric space, so it can be used to create a similarity matrix in non-metric settings, as is the case for moral foundations. Normalization needs to be done in a similar manner to cosine similarity.

**Hierarchy** While we do not perform any experiments with it, we showcase theoretically how to use a label hierarchies for the Semantic F1 (Figure 8). In this case, the distance between two labels is their shortest distance in the hierarchy graph, potentially weighted by edge weights. We can use that similarly to the Euclidean distance to derive similarities. This approach might be useful for settings with a hierarchical ontology, like music tagging (Smith et al., 2011), biological tasks, etc.

**Associative and Semantic Relations** A critical limitation of some of the aforementioned approaches lies in the distinction between associative and semantic similarity when deriving similarity matrices from embeddings or correlations. Word embeddings and co-occurrence statistics capture

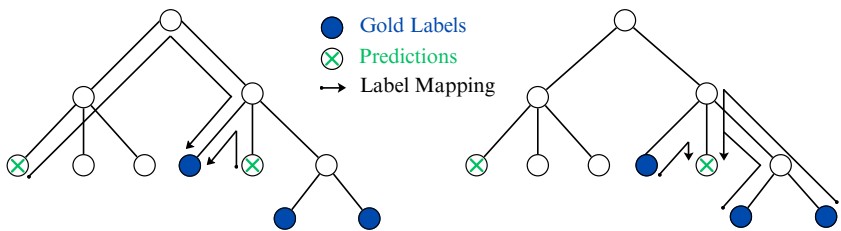

Figure 8: Example shortest paths in a label hierarchy for both steps of our algorithm.

distributional patterns, but this associative similarity may poorly represent the conceptual related-ness needed for meaningful evaluation. For instance, "doctor" and "patient" exhibit high embed-ding similarity due to frequent co-occurrence, yet confusing these labels in a medical classification task should not receive substantial partial credit. When we suggest using embedding distances or training correlations to construct similarity matrices, we essentially delegate core evaluation phi-losophy to statistical patterns rather than meaningful conceptual relationships. This approach may succeed when distributional and semantic similarity align, as is the case with ontologies fuzzy class boundaries, where co-occurrence does reflect semantic similarity, but also partially reflects the con-fusion of the concepts by humans due to that similarity, but fails systematically when they diverge, a common occurrence in specialized domains where technical precision matters more than linguistic association.

An interesting prospect is a per-item or per-annotator similarity matrix. Some label relationships could conceivably slightly change between examples, depending on the stimulus. For an even more accurate evaluation, the similarity could be adjusted based on gold information about the labels of the stimulus. For example, using annotator confusion for each example could inform the similarity matrix. However, using the stimulus itself to modify the similarity matrix, at least from the user perspective, would be circular: if we knew the ground-truth way to modify the similarity matrix appropriately, then we would know how to perform the classification itself. From the data distrib-utor's side, this is an interesting application for closed evaluations. By not sharing the modified similarity matrix, the distributor is not leaking additional information about the test labels, and can use information from the distribution of labels to modify the similarity matrix in a principled way.

## C  INTERPRETATION OF SIMILARITIES AND THE IDENTITY MATRIX

Care is required when interpreting similarity matrices, particularly in relation to the identity matrix.

**Metric-based similarities.**  When similarity is derived from distances in a metric space (e.g., Eu-clidean), the similarity matrix approaches the identity as distances grow large ($D \to +\infty$). In practice, sufficiently separated categories behave as orthogonal, and the metric smoothly collapses to standard F1 when $S = I$.

**Cosine and correlation similarities.**  However, in the case where values outside the range of $[0, 1]$ happen to be used, like correlations or the cosine similarity, then the interpretation of the similarity values, including the similarity of zero in the identity matrix, is trickier. That is because a normalized value of $0$ for correlation corresponds to anticorrelated labels, not uncorrelated labels. Nevertheless, for practical purposes, setting $S$ to be $0.5$ outside the diagonal, the mathematically appropriate value, produces the same or similar ranking between predictions, making it effectively equivalent to $S = I$, which has the nice property of collapsing to the standard F1 score.

**Design choices.**  We avoid allowing $S_{a,b} \in [-1, 1]$ directly, as this would require redesigning the harmonic mean to handle negative values. Instead, our normalization choices preserve the desirable property that Semantic F1 reduces exactly to standard F1 under $S = I$, while still accommodating richer similarity structures in practice.

In summary, similarity matrices can be constructed from metric distances, embeddings, correlations, or hierarchies. Each choice encodes different assumptions about the label space, but by design Semantic F1 always falls back to hard F1 when the matrix is the identity, ensuring interpretability and robustness.

## D    EDGE CASES AND VARIANTS

In this section, we elaborate on the formulas presented in the main text, presenting how we handle edge cases. First, for the BestMatch algorithm from Eq. 1, the complete formula is

$$\text{BestMatch}(A, B, S) = \begin{cases} \frac{1}{|A|} \sum_{a \in A} \max_{b \in B} S_{ab} & \text{if } A \neq \emptyset, B \neq \emptyset, \\ 1 & \text{if } A = \emptyset, B = \emptyset, \\ 0 & \text{otherwise.} \end{cases} \tag{14}$$

For the pointwise Semantic F1 score in Eq. 4, the full formulation is:

$$\text{SeF1}_i = \begin{cases} \frac{2 \cdot \text{Precision}_i^s \cdot \text{Recall}_i^s}{\text{Precision}_i^s + \text{Recall}_i^s} & \text{if } \text{Precision}_i^s + \text{Recall}_i^s \neq 0, \\ 0 & \text{otherwise.} \end{cases} \tag{15}$$

For the micro variant, we will first define precision and recall explicitly based on the global counts from Eq. 7, 8, 9:

$$\text{Precision}_{\text{micro}}^s = \frac{TP}{TP + FP} \tag{16}$$

$$\text{Recall}_{\text{micro}}^s = \frac{TP}{TP + FN}. \tag{17}$$

Similar to the pointwise Semantic F1 score, the edge cases of the Micro Semantic F1 score are:

$$\text{SeF1}_{\text{micro}} = \begin{cases} \frac{2 \cdot \text{Precision}_{\text{micro}}^s \cdot \text{Recall}_{\text{micro}}^s}{\text{Precision}_i^s + \text{Recall}_{\text{micro}}^s} & \text{if } \text{Precision}_{\text{micro}}^s + \text{Recall}_{\text{micro}}^s \neq 0, \\ 0 & \text{otherwise.} \end{cases} \tag{18}$$

For the Semantic Macro F1 score, the per class Semantic F1 score is:

$$\text{TP}_c = \sum_{i=1}^{n} \begin{cases} S_{M_{P,T}(c),c} & \text{if } c \in P_i \text{ and } M_{P,T}(c) \text{ exists} \\ 0 & \text{otherwise} \end{cases} \tag{19}$$

$$\text{FP}_c = \sum_{i=1}^{n} \begin{cases} 1 - S_{M_{P,T}(c),c} & \text{if } c \in P_i \\ 0 & \text{otherwise} \end{cases} \tag{20}$$

$$\text{FN}_c = \sum_{i=1}^{n} \begin{cases} 1 - S_{c,M_{T,P}(c)} & \text{if } c \in T_i \\ 0 & \text{otherwise} \end{cases} \tag{21}$$

### D.1    WEIGHTED SEMANTIC F1

We can extend Macro Semantic F1 score to use class support (frequency in ground truth) as weights:

$$\text{SeF1}_{\text{weighted}} = \frac{\sum_{c \in \mathcal{L}} w_c \cdot \text{SeF1}_c}{\sum_{c \in \mathcal{L}} w_c} \tag{22}$$

where $w_c = \sum_{i=1}^{n} \mathbf{1}[c \in T_i]$.

## E    EXTRA IMPLEMENTATION DETAILS

We use one Nvidia A100 to perform local inference with LLMs, and one NVIDIA RTX 6000 for training of Demux and Llama. Synthetic experiments were performed on the CPU. We train Demux exactly as described in the original paper Chochlakis et al. (2023a). We finetune Llama-3 1B with a new classification head with QLoRA (Hu et al., 2021) of rank 4 on KVQ. During different runs of LLM inference, we completely resample prompt examples.

We use exactly the same prompt format across all 6 LLMs across all tasks besides P4G, appropriately changing the instructions. An example on GoEmotions is:

> Classify the following inputs into none, one, or multiple the following emotions per input: joy, optimism, admiration, surprise, fear, sadness and anger. Output exactly these emotions and no others.
>
> Input: "Can I speak to the Suns' manager?"
> {"label": ["surprise"]}

For P4G, the corresponding presentation of the conversations in the prompts take the following format:

> A multi-turn conversation will be presented to you in the following format:
> CONVERSATION:
> ```
> conversation goes here
>
>
> ```
>
> Evaluate the last turn only for the expressed emotion of the speaker. This is important; do not take into account the emotions expressed previously in your assessment, but only to contextualize the last turn. Choose none, one, or multiple of the following emotions: anger, anticipation, disgust, fear, joy, love, optimism, pessimism, sadness, surprise, trust. Pick from these emotions only. Pick emotions that are plausible under some interpretation of the stimulus, but the emotions should make sense together as a group.
> The response should strictly follow this format:
>
> EMOTIONS: list of emotions
>
> for example 'EMOTIONS: anger, sadness', or 'EMOTIONS: optimism, love, joy'.
>
> CONVERSATION:
> ```
> Turn 0: Good Evening
> Turn 1: Hello there. how are you?
> Turn 2: I am doing well! How are doing today?
> Turn 3: I am doing pretty well. thanks for asking!
> Turn 4: I"d like to tell you about a great program I am working on! Have you ever heard of Save the Children?
> Turn 5: I may have in passing, but could you tell me more information about it?
> ```
>
> EMOTIONS: anticipation.

## F    ADDITIONAL RESULTS

Here, we present complete results that have been delegated to the appendix due to space constraints.

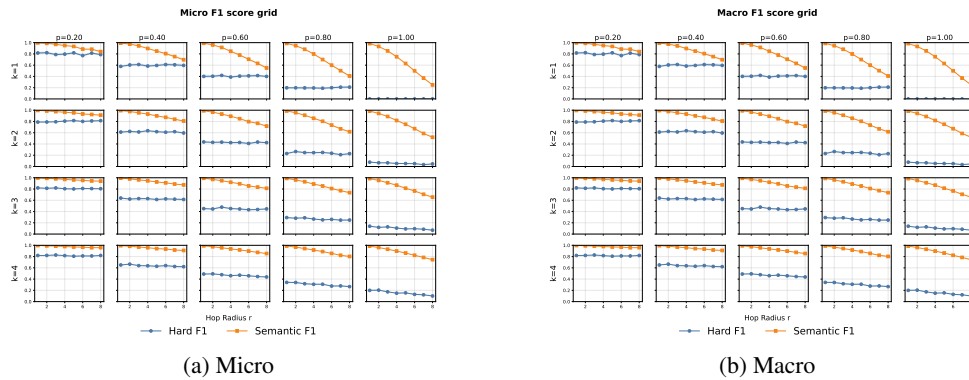

(a) Micro        (b) Macro

Figure 9: Hard vs Semantic F1 score across number of labels $k$, perturbation probability $p$, and hop radii $r$.

## F.1 SYNTHETIC STUDY A

We first present how micro and macro F1 scores, hard and semantic, vary with number of labels $k$, perturbation probability $p$, and hop radius $r$ in Figure 9. Conclusions reflect those in the main text. We also present the distribution of differences for macro and samples F1 scores in Figure 10, with similar trends shown as in the main text.

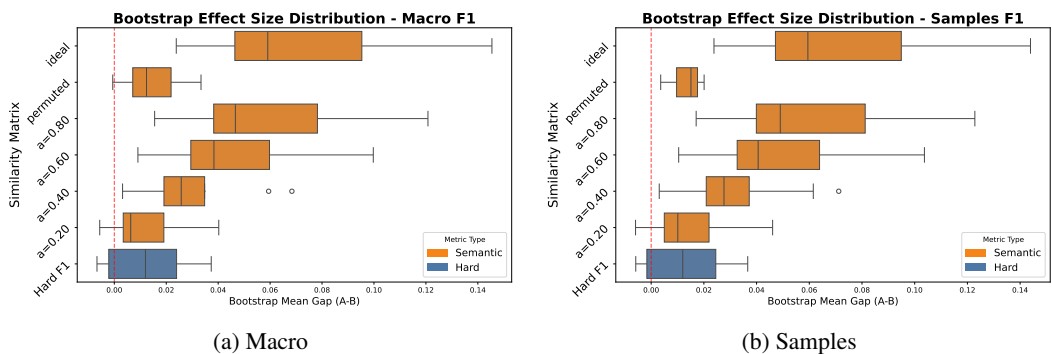

(a) Macro        (b) Samples

Figure 10: Distribution of differences in F1 between Near and Far predictions, aggregated across perturbation probabilities, number of labels, and radii.

We also report Kendall's $\tau$ in Figure 11, where we quantify the degree of monotonicity in the metrics. Because we expect decreasing trends, we show negative Kendall's $\tau$, meaning that higher values show a higher monotonic (decreasing) trend. We see that hard metrics do not reliably have a decreasing trend, with many instances showing an increasing trend with semantically less related predictions, or a constant trend.

Finally, we also report the correlation between the rankings of the various classifiers (defined by hop radius $r$) across all the similarity matrix noise levels (defined by $\alpha$). We aggregate results across all $p > 0$ and $k$. Results are shown in Figure 12. This further verifies that down to $\alpha = 0.4$, noise is still not sufficient to really to obscure the information we can derive for our classifiers from Semantic F1.

## F.2 SYNTHETIC STUDY B: BIMODAL HEURISTIC VS. FINE-GRAINED PREDICTOR

**Motivation** We test whether Semantic F1 properly penalizes "coarse correctness" when a predictor captures the correct *mode* but misses fine-grained labels. For example, some predictors might solely rely on the sentiment of a stimulus rather than finding the specific emotion, predicting the same positive or negative emotions depending on the identified sentiment. Intuitively, we would want a more fine-grained classifier, which might still make small errors, to be preferred by our met-

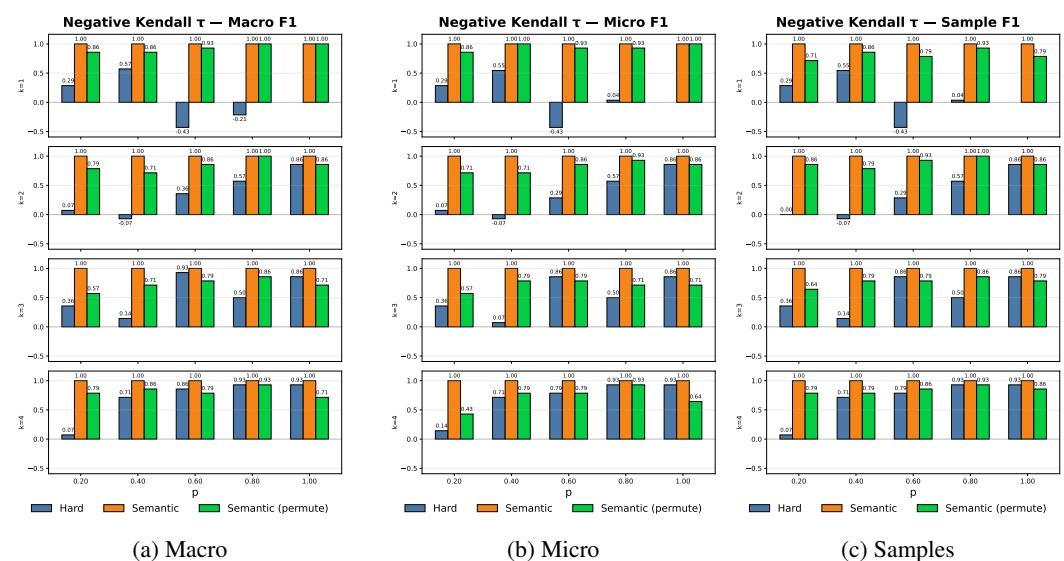

(a) Macro        (b) Micro        (c) Samples

Figure 11: Negative Kendall's $\tau$ (higher is better) across settings.

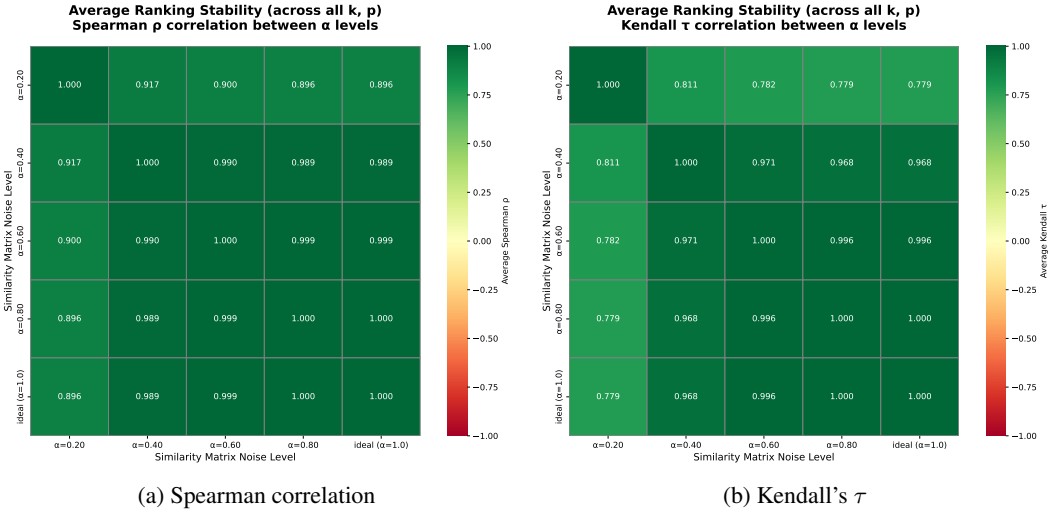

(a) Spearman correlation          (b) Kendall's $\tau$

Figure 12: Micro F1 ranking stability: Spearman correlation and Kendall's $\tau$, aggregated across $p$ and $k$, of the ranking of classifiers defined by their hop radius $r$.

rics. We construct a bimodal synthetic setting atop a circular label geometry (again, see the individual components in Figure 14) and compare Hard vs. Semantic F1 under controlled mode-selection behavior.

**Setup.** To extend the setup, we define two latent modes on the circle (positive and negative) using von Mises-like weights with concentration $\kappa$: $w_{\text{pos}}(j) \propto e^{\kappa \cos \theta_j}$ and $w_{\text{neg}}(j) \propto e^{\kappa \cos(\theta_j - \pi)}$. To generate gold sets, we first choose a mode with imbalance ratio $\rho \in \{0.25, 0.5, 0.75\}$, then sample $k \in \{2, 3\}$ labels from that mode's distribution. We evaluate prototype-based predictors that operate at the mode level: (i) **Prototype-Bimodal**: Determine the gold's dominant mode by summing mode weights over gold labels; predict that mode's $m \in \{2, 3, 4\}$ prototype labels with probability $q \in \{0, 0.2, 0.4, 0.6, 0.8, 1\}$, otherwise predict the opposite mode's prototypes. (ii) **Prototype-Within-Mode**: Choose the gold's mode with probability $q$ then sample $k$ labels from a distribution peaked on that mode's prototypes (tail controlled by $\beta$), otherwise from the opposite

mode. (iii) **Baselines**: Perturbation predictors from Study A.1 for reference, with $p = 1.0$ (exact matches only by chance). We use $n{=}24$ labels and $1000$ examples per configuration.

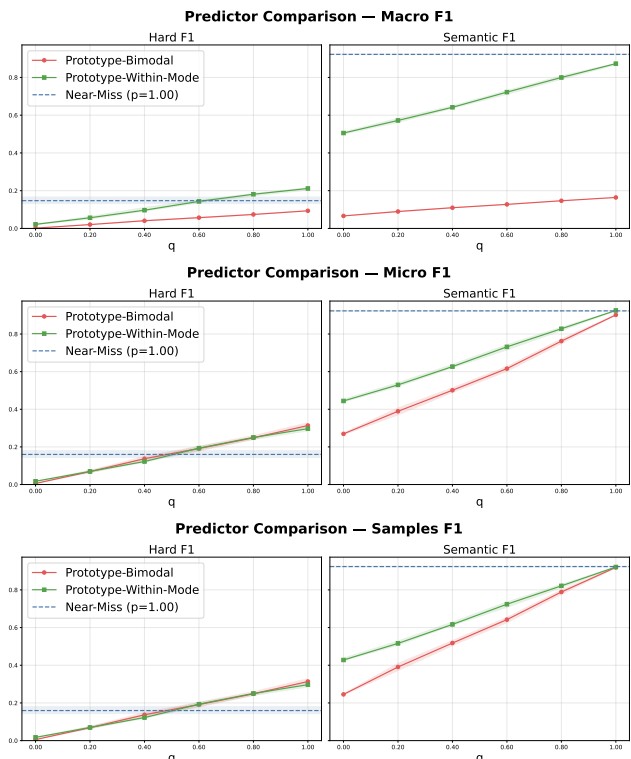

Figure 13: Comparison of bimodal heuristic classifiers compared to near-miss "intelligent" classifier.

**Metrics and statistics.** We report Hard F1 (micro/macro/samples) and Semantic F1 (micro/macro/samples). For probability sweeps, we plot metric vs. $q$ with $B = 20$ bootstrap 95% CIs for the prototype predictors and overlay a horizontal reference for a perturbation baseline.

**Results.** We compare the performance of the heuristic classifiers across different correct mode probability $q$ with the performance of the near miss, intelligent classifier in Figure 13. We see that across all different metrics, the heuristic classifiers outperform or are comparable to the intelligent classifier in hard scores, but that is not necessarily the case with the semantic scores. This is especially the case with samples F1. We conclude, therefore, that hard metrics cannot distinguish between mode heuristics effectively, whereas semantic metrics have the ability to.

### F.3   SYNTHETIC STUDY C: NON-METRIC SPACES

**Motivation** We stress-test Semantic F1 on a union-of-manifolds label space to assess whether geometry-aware similarities continue to distinguish near vs. far perturbations when labels lie on mixed structures. By contrasting ideal similarities with misaligned proxies (namely, a misguided assumption of a metric space), we show how naive metric choices can collapse Semantic F1 back toward hard F1 behavior, providing valuable guidance for practitioners. While the specific geometry is indeed synthetic (no immediate real world parallel)

**Setup.** Using the ring structure from Study A, we create a disjoint manifold as the union of ring structures, shown in Figure 14. Within each ring structure, the space is metric, but the disjoint manifold structure dictates that this is not so across manifolds. We modify the near-miss and far-miss predictors to include an additional parameter, $p_{jump}$. This quantifies the probability that a prediction will hop between manifolds. This is on top of $p$, which dictates the probability of a hop from gold labels to predictions. For simplicity and without loss of generality, we assume symmetric structures

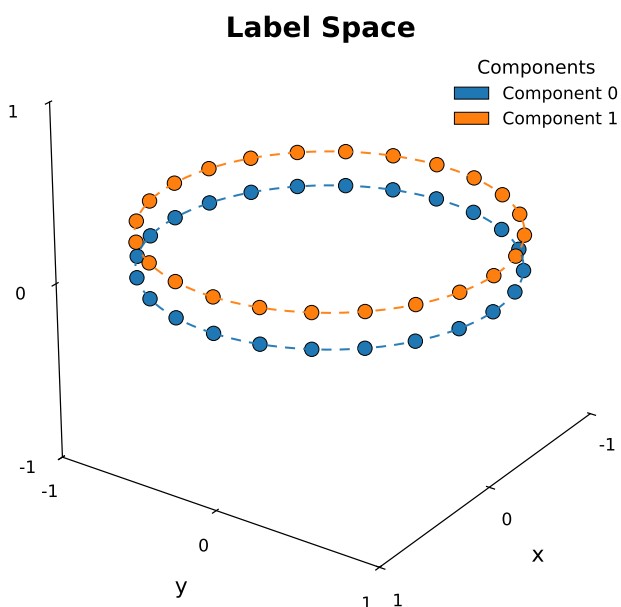

Figure 14: Non-metric label space: Disjoint union of manifolds

between rings, creating pairs of labels from each manifold. Similarity from a label of the other manifold is maximum at its pair label, and decays identically to the within-manifold behavior. Each configuration evaluates Semantic F1 under three matrices: (i) **Ideal**: geometry-derived similarities that respect manifold connectivity. (ii) **Permuted**: row-permuted control mirroring Study A.1 to break structure. (iii) **Deceptive Euclidean**: $1/(1 + \|.\|_2)$ over points in three-dimensional space, assuming unit radius and a distance of $0.2$ between rings, causing parallel manifolds to appear adjacent.

**Metrics and statistics.** We report hard F1 alongside Semantic F1 for each similarity matrix. We sweep $k \in \{1, 2, 3, 4\}$, $p = 1$, near radii $\{1, 2, 3, 4\}$, far radii $\{4, 5, 6, 7\}$, and $p_{jump} \in \{0, 0.2, 0.4, 0.6, 0.8, 1\}$. We present a grid of $k$-$p_{jump}$ plots with varying radii. We set $n = 48$ with each ring having 24 labels, making each identical to synthetic study A.

**Results.** In Figure 15, we see that the permuted and the deceptive similarity matrices, as well as the hard F1 score are invariant to the increase of $p_{jump}$. In addition, the permuted similarity matrix and hard F1 score are invariant to the increase of the hop radius, as we saw in study A as well (§4.3.1). In contrast, we see that Semantic F1 decreases linearly with hop radius when the hops happen in the same manifold, whereas it is relatively insensitive to the hop radius when most errors land in a different manifold (as all the labels are considered very distant in semantic space). It also decreases linearly with $p_{jump}$, as desired. Moreover, from Figure 16, we see that the moderately misspecified similarity matrix with $\alpha = 0.5$ also shows the same separability with the ideal matrix. Additionally, we see that the deceptive matrix shows much higher separability between predictors, when that should not be the case given the similarly low scores of cross-manifold jumps. Hard F1 and permuted similarity matrix show the same behavior, clustered around 0.

F.4 SYNTHETIC STUDY D: EMPIRICAL COMPARISON TO BASELINES

**Motivation.** We compare Semantic F1 to previous similarity-based single-step metrics, in particular with semantic recall, semantic precision motivated by the work of Turki et al. (2020), and the extended Hungarian, which we develop as a strong baseline and present in §A. We show how their failure cases can manifest in plausible settings and affect their ability to capture nuances in predictive behavior, in contrast to the Semantic F1 score.

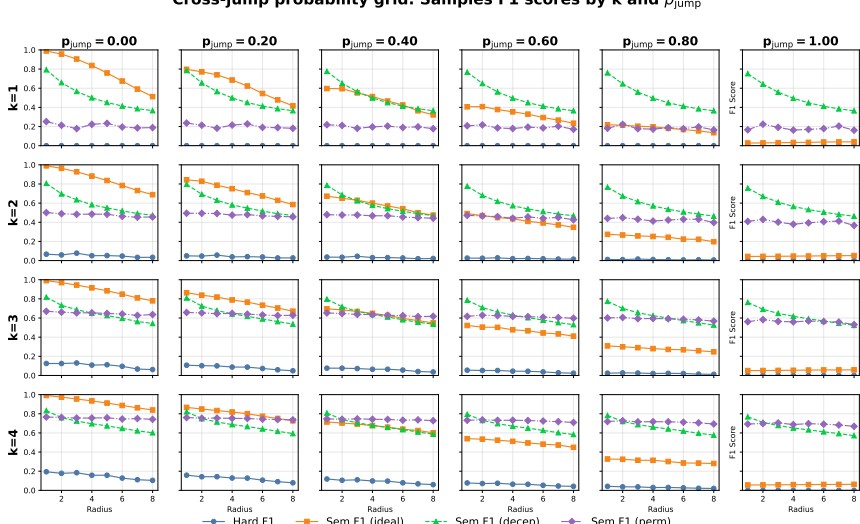

Figure 15: Hard vs semantic samples F1 score across number of labels $k$, cross-manifold hop probability $p_{jump}$, and hop radii $r$. Semantic F1 is also shown with a deceptive Euclidean-based similarity matrix, and a permuted similarity matrix.

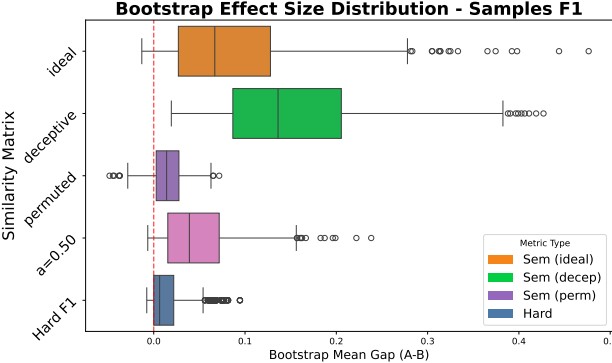

Figure 16: Distribution of differences in F1 between Near and Far predictions, aggregated across across-manifold jump probabilities, number of labels, and radii.

**Setup.** Using the ring geometry from synthetic study A, gold and predicted label sets are sampled around mode centers via softmax-weighted cosine similarity. We vary the number of gold labels $k$, a hop/perturbation probability $p$, the frequency that gold or predicted labels are bimodal $p_b$, and, in one scenario, the number of predicted labels. Three stress tests are run: (i) *Precision stress test (bimodal gold, unimodal predictor).* A proportion $p_b$ of examples have bimodal gold labels (two opposite ring modes). The predictor remains unimodal and, on bimodal examples, only hops around one mode in label space with probability $p$. We compare Semantic F1 to semantic precision as the frequency of bimodality varies, and show recall for reference. (ii) *Recall stress test (unimodal gold, bimodal predictor).* Gold is always unimodal. The predictor is bimodal with controlled frequency and, when bimodal, predicts $k/2$ labels from the gold mode and $k/2$ from another mode; otherwise it locally hops around the gold mode. We compare Semantic F1 to semantic recall across the predictor's bimodality frequency, and show precision for reference. (iii) *Hungarian stress test (bimodal gold, unimodal predictor, varying prediction counts).* Gold may be uni- or bimodal as above. The predictor outputs a controlled number of labels centered on one mode, sweeping the number of predictions while holding $k$ fixed. We compare Semantic F1 to the extended Hungarian score. All

comparisons use sample Semantic F1 as the most natural comparison to the sample-based Hungarian score.

**Metrics and statistics.** For each configuration (values of $k$, bimodality frequency, $p$, and, when applicable, number of predictions), we generate synthetic datasets of 1000 examples per configuration and report means. Plots aggregate over nuisance variables (e.g., averaging across $k$ and $p$ where appropriate) to show mean curves for Semantic F1 and the relevant baseline. We choose $n \in \{96, 192\}$ (to allow us to scale the number of predictions within a single space to 17 without expanding to other subspaces in label space, and therefore get smoother curves), $k = 2$, and predictive temperature of $0.1$ for the Hungarian comparison, and $n = 96$, $k \in \{6, 8, 10\}$, $p = 1$, a hop radius of 2, and gold sampling temperature of $0.05$.

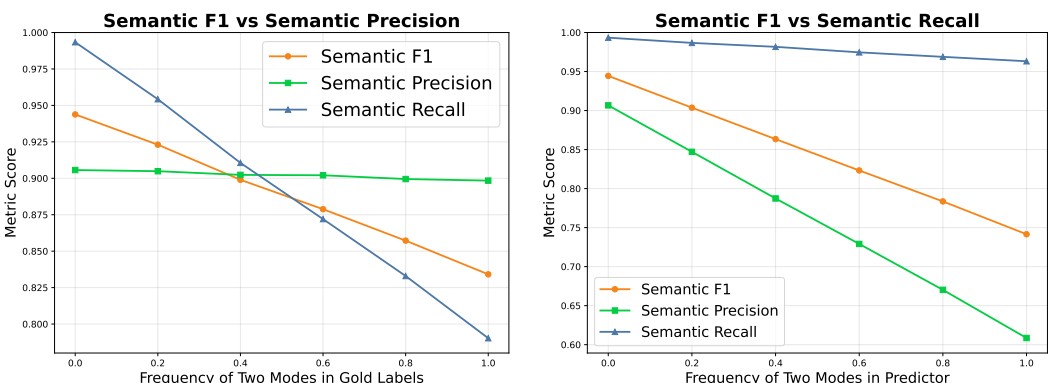

Figure 17: Failure modes when using only semantic precision (left) and only semantic recall (right).

**Results.** Figure 17 show that, as expected, *precision* fails to take into account the under-coverage of the label space when that becomes bimodal, whereas *recall* does not penalize over-prediction when the predictor becomes more and more bimodal in a unimodal label space. Semantic F1 properly scales down in both cases. It is worth noting that flipping gold and predicted labels in each setting flips the metric that has a failure mode, yet Semantic F1 remains the same. In Figure 18 (figures are identical for $n = 96$ and $192$, so we show only the latter.), we see how the Hungarian algorithm rewards bad predictors. As we increase the rate of bimodal label spaces, we see that the performance of a unimodal predictor does not decrease, as measured by the Hungarian score. This happens because the predictor can predict the neighbors in its captured mode to artificially increase its score with the Hungarian algorithm, as can be seen by the large increases in performance when the predictor adds more and more predictions in the same region. In contrast, the Semantic F1 seems much more moderate scaling when predictors increase within each setting, and, notably, performance degrades as the label space becomes more bimodal. This scaling behavior in Semantic F1, as opposed to the Hungarian algorithm, is a byproduct of the similarity matrix used and not the method itself. For instance, if we saturate the similarity to 0 faster as we move away from each label, for example by squaring or cubing it, the effect is eliminated for Semantic F1, but not for the Hungarian score, as can be seen in Figure 19.

### F.5 REAL STUDY A

Here, we present how the results for Llama-3 1B, as shown in the main text for Demux, in Figure 20, and how performance varies per threshold on the rest of the datasets, GoEmotion and MFRC, in Figures 21 and 22 respectively.

### F.6 REAL STUDY B

For completeness, we present the full correlations for all metrics for both datasets in Figures 23, 24 and 25. As noted before, the correlations to Semantic F1 metrics are at least as large as with the hard F1 scores in all settings, a good indicator that semantic metrics are more ecologically valid in problems with interrelated, fuzzy labels. In Tables 2 and 3, we see that this is not in general the case

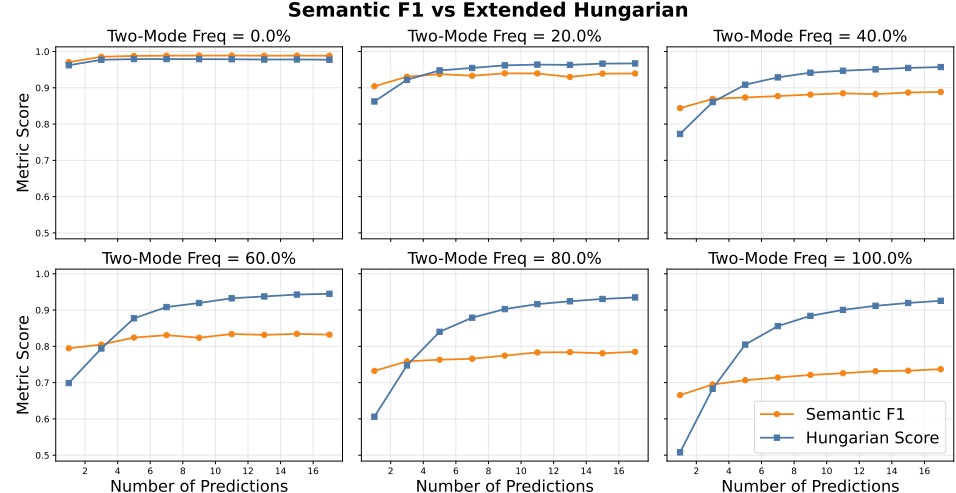

Figure 18: Semantic F1 vs Hungarian: Hungarian drowns out missed label space by predicting more and more labels around another label mode.

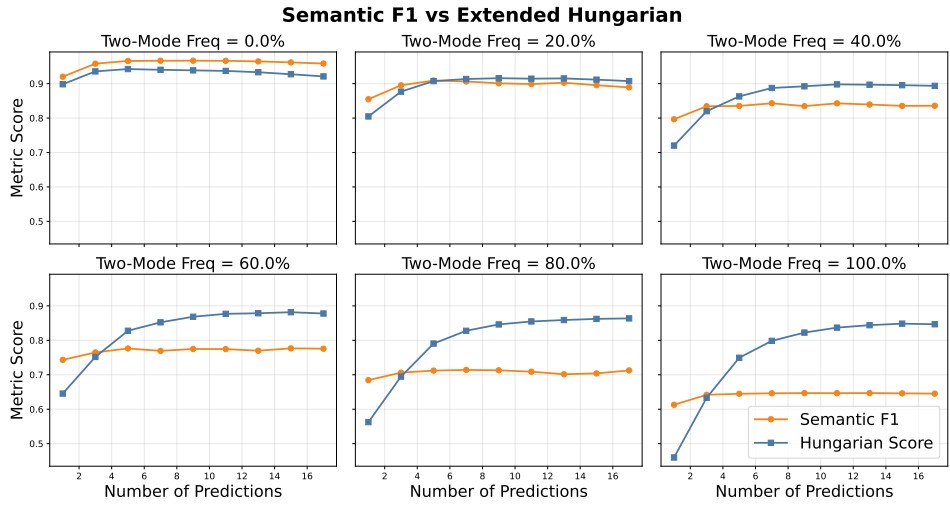

Figure 19: Semantic F1 vs Hungarian with *cubed* similarity matrix: Hungarian still scales with more predictions around one label mode.

for individual semantic components, and that in a general setting, both of them need to be combined to the Semantic F1 to provide reliable results. It is interesting to see that correlation with Semantic Precision is low; it indicates that under-coverage is an important property for emotional downstream tasks.

| | Precision | Recall | Hard F1 |
|---|---|---|---|
| Samples | 0.348 | 0.754 | |
| Micro | 0.232 | 0.899 | 0.696 |
| Macro | 0.522 | 0.986 | |

Table 2: Correlations of P4G and SemEval

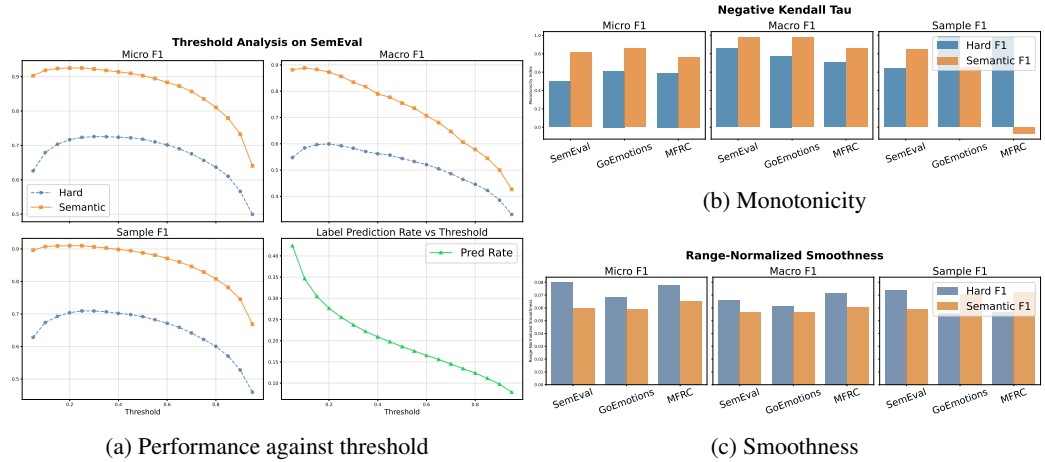

(a) Performance against threshold

(b) Monotonicity

(c) Smoothness

Figure 20: Threshold analysis on Llama-3 1B

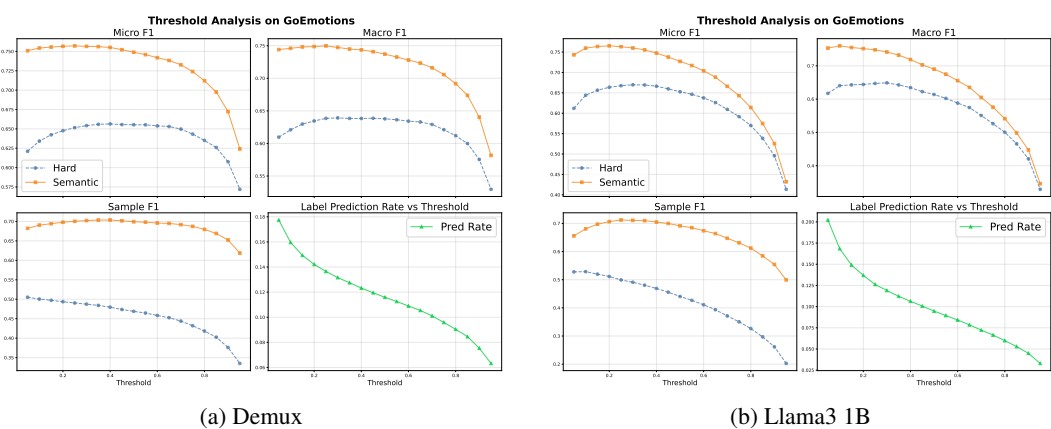

(a) Demux

(b) Llama3 1B

Figure 21: Semantic and hard F1 scores across probability thresholds on GoEmotions.

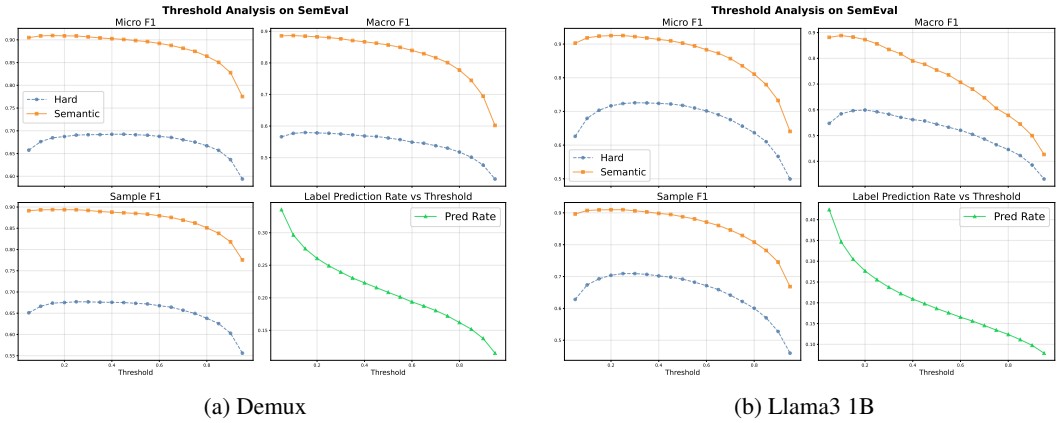

(a) Demux

(b) Llama3 1B

Figure 22: Semantic and hard F1 scores across probability thresholds on MFRC.

## F.7 REAL STUDY C

We present detailed results on early stopping for all subjective multi-label datasets, in Figures 26, 27 and 28. Aggregated results for these datasets were already shown in Table 1. As noted, we see that semantic early stopping leads to significant gains even in hard metrics, like Jaccard Score in MFRC.

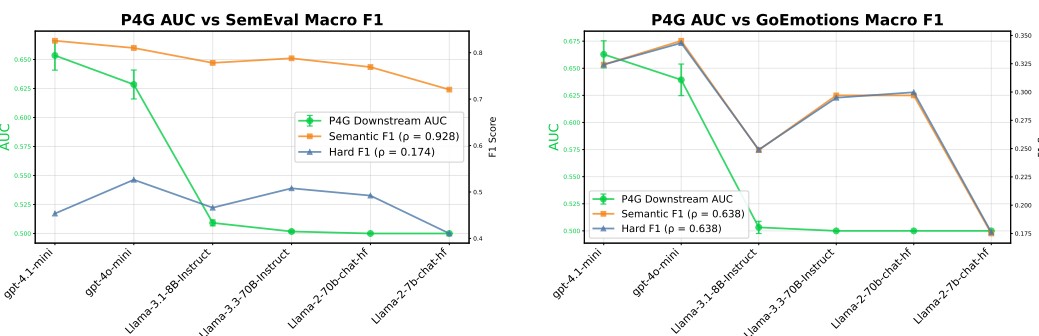

(a) Correlation between F1s and downstream with SemEval

(b) Correlation between F1s and downstream with GoEmotions

Figure 23: Ecological validity study results comparing macro Semantic F1 vs hard F1 correlation with downstream task performance across different emotion datasets

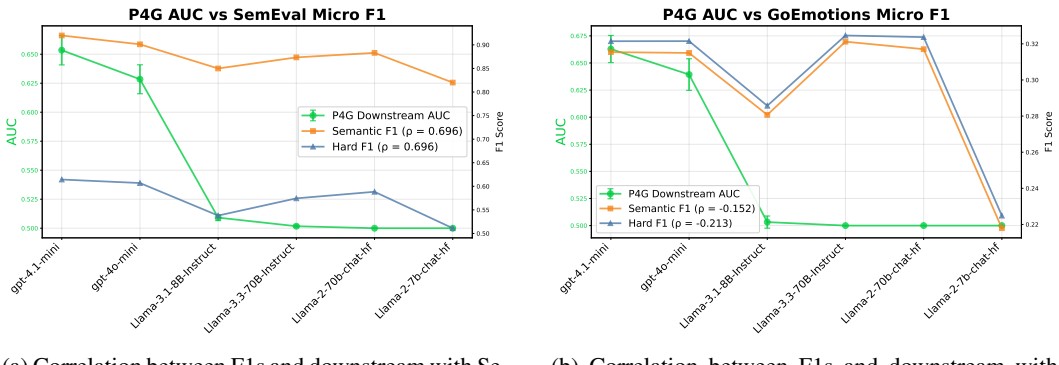

(a) Correlation between F1s and downstream with SemEval

(b) Correlation between F1s and downstream with GoEmotions

Figure 24: Ecological validity study results comparing micro Semantic F1 vs hard F1 correlation with downstream task performance across different emotion datasets

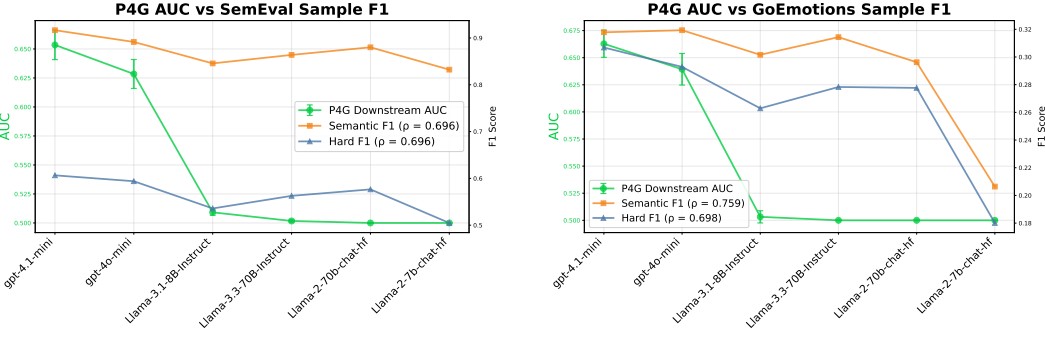

(a) Correlation between F1s and downstream with SemEval

(b) Correlation between F1s and downstream with GoEmotions

Figure 25: Ecological validity study results comparing samples Semantic F1 vs hard F1 correlation with downstream task performance across different emotion datasets

## F.8 REAL STUDY D: CONVERGENT VALIDITY

We use the following additional, objective datasets to evaluate how scaling correlates between objective and subjective tasks:

|         | Precision | Recall | Hard F1 |
|---------|-----------|--------|---------|
| Samples | 0.334     | 0.880  |         |
| Micro   | 0.575     | 0.941  | 0.698   |
| Macro   | 0.152     | 0.880  |         |

Table 3: Correlations of P4G and GoEmotions

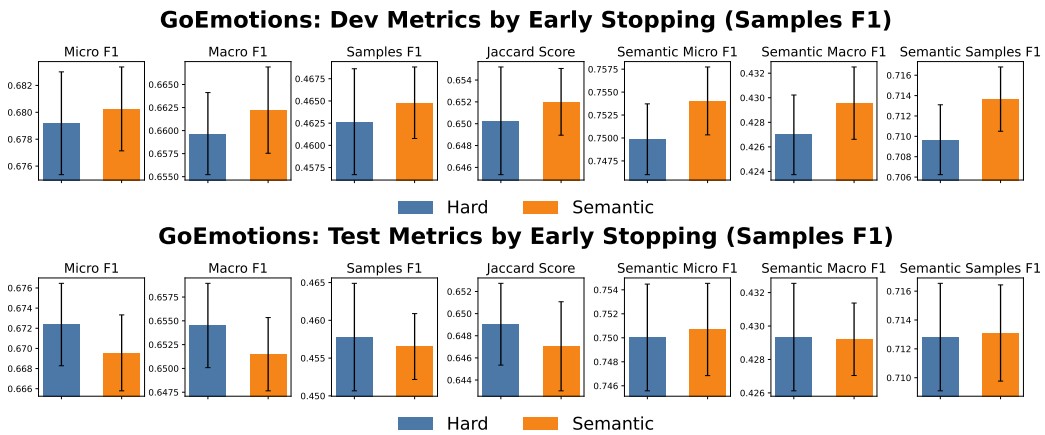

Figure 26: Performance comparison on MFRC across 6 F1 metrics and Jaccard Score when using hard or semantic samples F1 score as early stopping criterion. *: $p < 0.05$, **: $p < 0.01$.

Figure 27: Performance comparison on GoEmotions across 6 F1 metrics and Jaccard Score when using hard or semantic samples F1 score as early stopping criterion.

**MovieLens (Harper & Konstan, 2015)**   Objective multi-label movie genre prediction based on IMDB movie summaries.

**Boxes (Kim & Schuster, 2023)**   Objective multi-label entity tracking based on natural language description of "box" contents and "move" operations. Each box can contain none, one, or multiple objects. The dataset contains thousands of synthetic examples.

**TREC (Hovy et al., 2001; Li & Roth, 2002)**   Objective single-label question classification benchmark, which contains annotations for the type of information the question pertains to.

**Setup.**   We evaluate Semantic F1 on real datasets to test convergent validity: if Semantic F1 score is a better metric, then on subjective, fuzzy multi-label tasks it should align better with model capability in objective single-label and multi-label datasets compared to Hard F1 score.

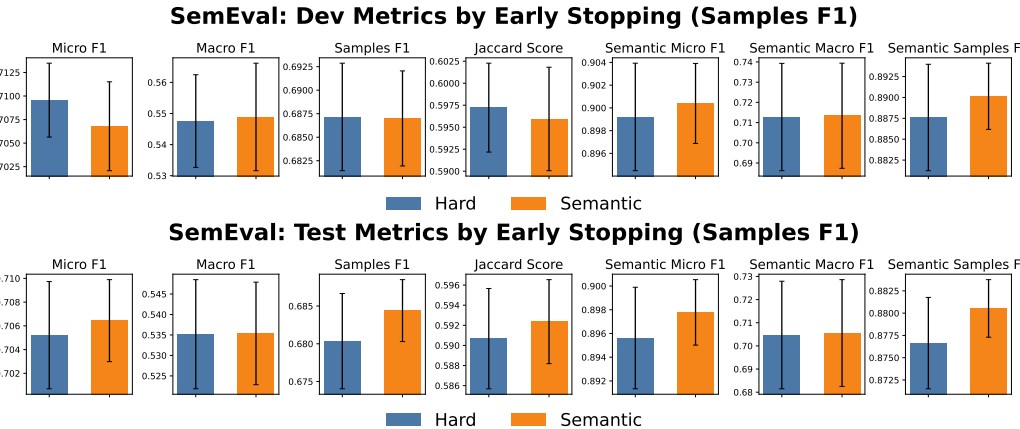

Figure 28: Performance comparison on SemEval across 6 F1 metrics and Jaccard Score when using hard or semantic sample F1 score as early stopping criterion.

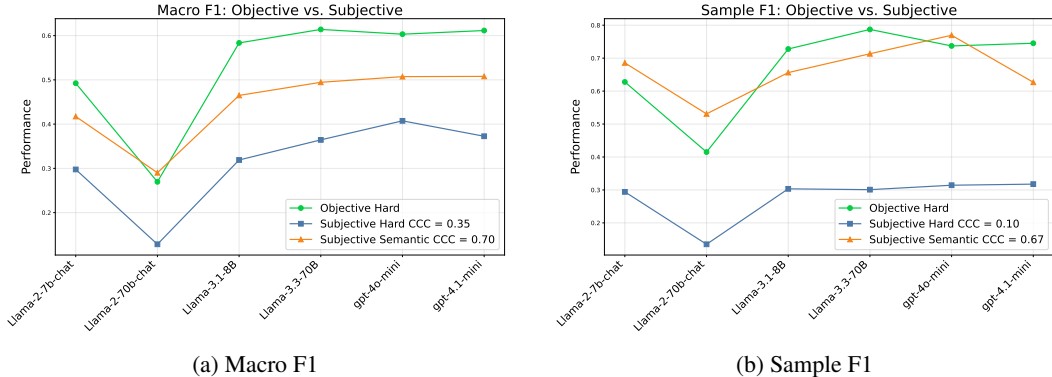

(a) Macro F1
(b) Sample F1

Figure 29: Semantic and hard F1 scores on subjective tasks correlated using CCC with hard F1 score performance in objective tasks.

**Metrics and statistics.** We compare the average performance on single-label and multi-label objective tasks, using hard F1 metrics, with the average performance on multi-label subjective tasks using semantic and hard F1 metrics (micro/macro/samples). We do not present micro F1, as it is not defined for single-label settings. We report the Concordance Correlation Coefficient (CCC) and the Spearman correlation of the metrics on the subjective tasks with the objective tasks. The CCC is employed in this scenario because performance is on the same scale for all corresponding metrics, and therefore tracking entails the element of matching the magnitude. Also, it was used as a complement to Spearman correlation, since we note that 95% CIs are large enough to make correlations and ranking volatile.

**Results.** Figure 29 shows the performance on objective and subjective tasks of 6 LLMs, and the CCC of the semantic and the hard subjective performance to the objective performance. We see that CCC is much higher using the Semantic F1 scores, suggesting that Semantic F1 tracks hard F1 score on objective tasks better than hard F1 on subjective tasks does. We note that Spearman correlation results are mixed; semantic macro F1 has $\rho = 0.83$, beating the $\rho = 0.77$ of hard macro F1, but hard sample F1 score beats semantic sample F1 with a Spearman correlation of $\rho = 0.66$ compared to $\rho = 0.49$. For macro F1, we also present all the objective-subjective dataset pairs in Figure 30.

## G  PSEUDOCODE

In this section, we present the pseudocode for BestMatch, Samples, Micro, and Macro Semantic F1 scores in Algorithms 1, 2, 3 and 4 respectively.

---

**Algorithm 1** BestMatch

---

**Require:** Label set $A$, Label set $B$, Similarity matrix $S$
  similarityScores $\leftarrow$ []
  $M_{A,B} \leftarrow \{\}$                                         // label pairs hashmap
  **for** each $a \in A$ **do**
    $b \leftarrow \arg\max_{x \in B} S_{a,x}$
    $M_{A,B}[a] \leftarrow b$
    similarityScores.append($S_{a,b}$)
  **end for**
  $\bar{s} \leftarrow \frac{1}{|A|} \sum_{s \in \text{similarityScores}} s$            // arithmetic mean of similarities
  **return** $\bar{s}$

---

**Algorithm 2** Samples Semantic F1 Score

---

**Require:** Predicted sets $\{P_1, P_2, \ldots, P_n\}$, True sets $\{T_1, T_2, \ldots, T_n\}$, Similarity matrix $S$
  F1 $\leftarrow$ 0                                     // accumulator for F1 scores
  **for** $i = 1$ to $n$ **do**
    prec $\leftarrow$ BestMatch($P_i, T_i, S$)         // Compute semantic precision
    rec $\leftarrow$ BestMatch($T_i, P_i, S$)          // Compute semantic recall
    $F1 \leftarrow F1 + \frac{2 \cdot \text{prec} \cdot \text{rec}}{\text{prec} + \text{rec}} \cdot \frac{1}{n}$     // Compute harmonic mean for F1
  **end for**
  **return** $F1$                          // arithmetic mean of F1 scores

---

**Algorithm 3** Micro Semantic F1 Score

---

**Require:** Predicted sets $\{P_1, P_2, \ldots, P_n\}$, True sets $\{T_1, T_2, \ldots, T_n\}$, Similarity matrix $S$
  Initialize TP $\leftarrow$ 0, FP $\leftarrow$ 0, FN $\leftarrow$ 0          // global semantic counts
  **for** $i = 1$ to $n$ **do**
    Compute forward pairs: $F_i \leftarrow \{(p, \arg\max_{t \in T_i} S_{tp}) : p \in P_i\}$
    Compute reverse pairs: $R_i \leftarrow \{(t, \arg\max_{p \in P_i} S_{tp}) : t \in T_i\}$
                                     // Accumulate semantic true positives
    **for** each $p \in P_i$ **do**
      **if** $p$ has forward match $t^* \in T_i$ **then**
        TP $\leftarrow$ TP $+ S_{t^*p}$
      **end if**
    **end for**
                                       // Accumulate semantic false positives
    **for** each $p \in P_i$ **do**
      **if** $p$ has forward match $t^* \in T_i$ **then**
        FP $\leftarrow$ FP $+ (1 - S_{t^*p})$
      **else**
        FP $\leftarrow$ FP $+ 1$
      **end if**
    **end for**
                                      // Accumulate semantic false negatives
    **for** each $t \in T_i$ **do**
      **if** $t$ has reverse match $p^* \in P_i$ **then**
        FN $\leftarrow$ FN $+ (1 - S_{tp^*})$
      **else**
        FN $\leftarrow$ FN $+ 1$
      **end if**
    **end for**
  **end for**
                                       // Compute global precision and recall
  precision $\leftarrow \frac{\text{TP}}{\text{TP}+\text{FP}}$ (if TP $+$ FP $> 0$, else 0)
  recall $\leftarrow \frac{\text{TP}}{\text{TP}+\text{FN}}$ (if TP $+$ FN $> 0$, else 0)
  F1 $\leftarrow \frac{2 \cdot \text{precision} \cdot \text{recall}}{\text{precision}+\text{recall}}$ (if precision $+$ recall $> 0$, else 0)
  **return** F1

---

**Algorithm 4** Macro Semantic F1 Score

**Require:** Predicted sets $\{P_1, P_2, \ldots, P_n\}$, True sets $\{T_1, T_2, \ldots, T_n\}$, Similarity matrix $S$, Label set $\mathcal{L}$

   Initialize $\text{TP}_c \leftarrow 0$, $\text{FP}_c \leftarrow 0$, $\text{FN}_c \leftarrow 0$ for all $c \in \mathcal{L}$      // per-class semantic counts

   Initialize $\text{support}_c \leftarrow 0$ for all $c \in \mathcal{L}$      // class frequencies

   **for** $i = 1$ to $n$ **do**

      Compute forward pairs: $F_i \leftarrow \{(p, \arg\max_{t \in T_i} S_{tp}) : p \in P_i\}$

      Compute reverse pairs: $R_i \leftarrow \{(t, \arg\max_{p \in P_i} S_{tp}) : t \in T_i\}$

      **for** each class $c \in \mathcal{L}$ **do**

        **if** $c \in T_i$ **then**

          $\text{support}_c \leftarrow \text{support}_c + 1$

        **end if**

                                        // Update semantic true positives

        **if** $c \in P_i$ and $c$ has forward match $t^* \in T_i$ **then**

          $\text{TP}_c \leftarrow \text{TP}_c + S_{t^*c}$

        **end if**

                                          // Update semantic false positives

        **if** $c \in P_i$ **then**

          **if** $c$ has forward match $t^* \in T_i$ **then**

            $\text{FP}_c \leftarrow \text{FP}_c + (1 - S_{t^*c})$

          **else**

            $\text{FP}_c \leftarrow \text{FP}_c + 1$

          **end if**

        **end if**

                                        // Update semantic false negatives

        **if** $c \in T_i$ **then**

          **if** $c$ has reverse match $p^* \in P_i$ **then**

            $\text{FN}_c \leftarrow \text{FN}_c + (1 - S_{cp^*})$

          **else**

            $\text{FN}_c \leftarrow \text{FN}_c + 1$

          **end if**

        **end if**

      **end for**

   **end for**

                                        // Compute per-class F1 scores

   **for** each class $c \in \mathcal{L}$ **do**

      $\text{precision}_c \leftarrow \frac{\text{TP}_c}{\text{TP}_c + \text{FP}_c}$ (if $\text{TP}_c + \text{FP}_c > 0$, else 0)

      $\text{recall}_c \leftarrow \frac{\text{TP}_c}{\text{TP}_c + \text{FN}_c}$ (if $\text{TP}_c + \text{FN}_c > 0$, else 0)

      $\text{F1}_c \leftarrow \frac{2 \cdot \text{precision}_c \cdot \text{recall}_c}{\text{precision}_c + \text{recall}_c}$ (if $\text{precision}_c + \text{recall}_c > 0$, else 0)

   **end for**

                                  // Return macro average or weighted average

   **if** macro averaging requested **then**

      **return** $\frac{1}{|\mathcal{L}|} \sum_{c \in \mathcal{L}} \text{F1}_c$

   **else**

      $\text{total\_support} \leftarrow \sum_{c \in \mathcal{L}} \text{support}_c$

      **if** $\text{total\_support} > 0$ **then**

         **return** $\frac{\sum_{c \in \mathcal{L}} \text{F1}_c \cdot \text{support}_c}{\text{total\_support}}$

      **else**

         **return** $\frac{1}{|\mathcal{L}|} \sum_{c \in \mathcal{L}} \text{F1}_c$      // fallback to macro

      **end if**

   **end if**

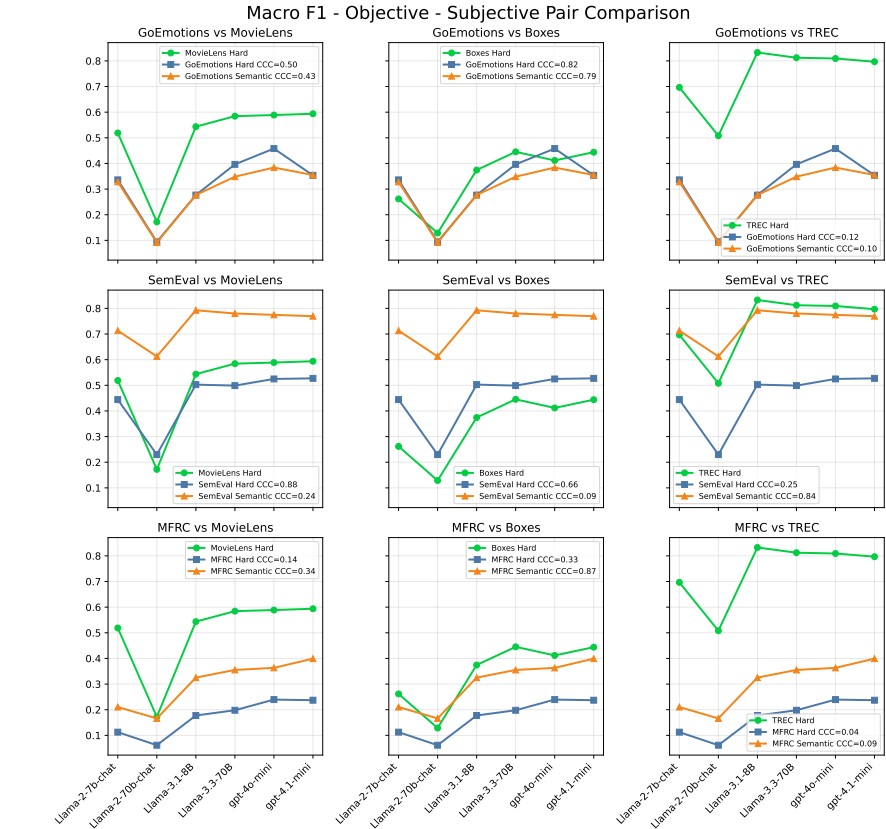

Figure 30: Semantic and hard macro F1 scores on subjective tasks correlated using CCC with hard F1 score performance in objective tasks, shown for every pair of objective-subjective dataset.

## H    LLM USAGE

OpenAI's ChatGPT was used to improve the prose in the paper. Antropic's Claude and OpenAI's ChatGPT were used internally as reviewers of the paper, helping us further improve the phrasing, while also suggesting experimental improvements. OpenAI's ChatGPT and Anthropic's Claude also assisted in the visualization of the results by providing the initial code for the plotting functions, and provided the backbone for the synthetic studies. All was manually checked and corrected by the authors.

