# OpenReview forum: "Semantic F1 Scores: Fair Evaluation Under Fuzzy Class Boundaries"
_ICLR.cc/2026/Conference — Submitted to ICLR 2026_

### Official Review · Reviewer_4Ly2 · 2025-10-20

**Soundness:** 3
**Presentation:** 3
**Contribution:** 2
**Rating:** 4
**Confidence:** 3

**Summary:**

The paper proposes a semantic F1 metric for multi-label classification. The proposed metric captures the ambiguity of class labels and can provide a fair and interpretable evaluation of the model's performance in the related tasks.

**Strengths:**

(1) The proposed metric effectively captures the fuzziness of class labels and the correlations between different class labels by integrating the semantic similarities among them.

(2) The calculation process of the proposed metric is simple and reasonable, and when the similarity of class labels degenerates to the identity matrix, the proposed criterion degenerates into the standard F1 criterion.

(3) A large number of systematic experiments have verified the effectiveness of the proposed criterion, including the fairness, interpretability, and robustness of the evaluation results.

(4) The experiments demonstrate that the proposed metric can provide guidance for early stopping, thereby achieving better generalization performance.

(5) The article also provides an accelerated calculation scheme for the proposed metric.

**Weaknesses:**

The proposed metric heavily rely on the quality of the class label similarity matrix.

**Questions:**

(1) Does the proposed metric have corresponding issues in the single-label classification task? Can the proposed metric be generalized to the single-label classification task?

(2) What are the connections and differences between the proposed metric and the cost-sensitive classification loss functions?

---

> ### Author Response · Authors · 2025-11-20
> **Point-by-point rebuttal**
>
> Dear reviewer, thank you for taking the time to review our paper and for the interesting questions. We again want to recognize your time and effort. Below find a point-by-point response to your comments:
>
> - Similarity matrix dependency: As discussed more extensively in our general comment, we alleviate this significantly, since our method maintains discriminative power even with 60% noise (Figures 2b, 15).
> - Single-label: Yes, Semantic F1 applies directly to single-label settings. We now mention this in the paper. We focused on multi-label because that's where existing metrics have the most significant gaps.
> - Cost-sensitive learning: The key difference is that cost-sensitive methods use 1-to-1 label weighting, which cannot handle sets of different sizes. Semantic F1's two-step formulation addresses this. We've added the discussion of Section 2.2 in the related work for more details.
>
> --Edited for clarity--

---

> > ### Comment · Reviewer_4Ly2 · 2025-11-26
> > **I maintained my score.**
> >
> > Thanks for the authors' reply. This has removed my doubts.
> >
> > I maintained my score.

---

> > > ### Author Response · Authors · 2025-11-27
> > >
> > > Many thanks for confirming that our answers have clarified the reviewer’s doubts. The updated paper also reflects these explanations. If the reviewer still has questions or doubts about our paper, causing them to maintain their score, we would appreciate the opportunity to elaborate and discuss further, as well as provide additional experiments.

---

### Official Review · Reviewer_x18W · 2025-10-25

**Soundness:** 3
**Presentation:** 3
**Contribution:** 2
**Rating:** 2
**Confidence:** 4

**Summary:**

The authors recognize the limitations of conventional F1 scores, which treat any non-exact label match as a complete failure, and introduce semantic F1 scores for multi-label classification tasks characterized by subjective or fuzzy class boundaries. The proposed metrics incorporate a domain-appropriate label similarity matrix to grant partial credit for semantically related predictions. Finally, the authors provides extensive empirical validation through multiple synthetic and real-world studies.

**Strengths:**

To address fuzziness of the semantic boundaries among labels, this paper introduces a two-step matching algorithm. It distinctly calculates semantic precision by mapping each prediction to its closest ground-trueh label and semantic recall by mapping each ground-truth label to its closest prediction. These components are then combined via harmonic mean to derive the final semantic F1 score, ensuring balanced penalties for both over-prediction and under-coverage in the semantic label space.

**Weaknesses:**

1. The novelty of this paper could be further strengthened. The proposed performance evaluation method based on semantic relevance appears quite similar to existing work on cost-sensitive learning; it would be helpful if the authors could clarify the distinctions between them.
2. The applicability of the semantic F1 score may require more discussion. Since it relies on a similarity matrix that is not entirely objective, the fairness of the final evaluation might benefit from further exploration.
3. The main contribution of the paper seems to be primarily centered around Equation (1), which seems relatively minor.

**Questions:**

1. Could you please clarify the key distinctions between the proposed evaluation methods and existing works in cost-sensitive learning? A more detailed discussion would help better highlight the novelty of your work.

2. Could you provide more justification for the use of the semantic F1 score? I have concerns regarding its fairness, as it relies on a non-objective similarity matrix. It would be helpful to discuss its potential limitations.

3. The paper's core contribution appears to be centered on Equation (1). Could you elaborate on whether there are other significant contributions, or discuss the impact and sufficiency of this primary contribution in greater depth?

---

> ### Author Response · Authors · 2025-11-20
> **Point-by-point rebuttal**
>
> Dear reviewer, we are grateful for the time and effort you devoted to reviewing our paper, we recognize that our extensive analysis might have made the review longer than average. Below find a point-by-point response to your comments:
>
> - Cost-sensitive learning: The key difference is that cost-sensitive methods use 1-to-1 label weighting, which cannot handle sets of different sizes. Semantic F1's two-step formulation addresses this directly. We've added this discussion to Section 2.2.
> - Similarity matrix dependency: As discussed more extensively in our general comment, we alleviate this significantly compared to previous work with the same dependency, since our method maintains discriminative power even with 60% noise (Figures 2b, 15).
> - Contributions: We appreciate the opportunity to clarify. The core contribution is not Equation 1 itself, but the two-step formulation that applies it bidirectionally to create interpretable semantic precision and semantic recall. This addresses failure modes of prior single-step approaches (over-prediction, under-coverage) while remaining backward-compatible with standard F1, as shown mathematically in the paper. We validated this across 8 datasets and with synthetic experiments, more extensively than prior semantic metrics. The formulation's simplicity is intentional; prior work introduced complexity without solving these failure modes. We've clarified contributions at the end of Section 1 (Introduction).
>
> --Edited for clarity--

---

### Official Review · Reviewer_hnNu · 2025-11-06

**Soundness:** 2
**Presentation:** 2
**Contribution:** 2
**Rating:** 2
**Confidence:** 2

**Summary:**

The paper provides a new metric, Semantic F1 Scores, suited for multi-label classification tasks, that does not excessively penalize failures in classification, but rather employs a similarity matrix to account for fuzzy ground truth labels or disagreement between annotators. The authors provide some experiments to evaluate the strength of their metric with respect to real-world and synthetic tasks.

**Strengths:**

The paper tackles an interesting and timely problem concerning the evaluation of predictive algorithms in multiclass tasks when the ground truth can be fuzzy or could represent some disagreement with respect to, for example, human annotators.

**Weaknesses:**

(Disclaimer: this is not exactly my area of expertise. I skimmed through the appendices, but I did not check them too deeply.)

- I found the problem formulation not very clear in its exposition. For example, I assumed the F1 score formulation would have been defined in a classical supervised learning setting (e.g., we have samples (x,y) where $X \sim P(X)$ and $y \sim P(Y)$ comes from unreliable annotators…), but here the author opted for a set-based formalization that I believe it is less clear.

- The BestMatch function (Equation 1) only considers the sets of predicted and “ground truth” labels. Thus, it considers all labels as globally related, thus, independently of the particular instance $x$ being evaluated. As far as I understood it, this is a critical issue. For example, in one context, _“irony”_ may be close to _“humor”_, in another to _“sarcasm”_ or _“criticism”_. A global embedding-based similarity cannot capture that nuance. Consequently, _SeF1_ may assign high partial credit even when the predicted label is semantically incompatible in context, simply because it is related in the global embedding space.

- The synthetic study (Section 4.3.1) is too abstract. It would have been better to connect it to a real synthetic classification task (e.g., by creating a simple multiclass classification problem). The same applies to the “perturbed predictors” and to the relevance of the study concerning misspecified similarities.

- No source code is available. It would have helped to better understand how to practically use the proposed metric (the pseudocode brings you only so far), and it would have solved my concern above.

- The tone of the paper sounds a bit like “overselling” at certain points. For example, in line 483, I do not believe that the proposed method provides a “more informative evaluation”, given that, as the authors also argue, if the similarity matrix $S$ is misspecified, then you can basically obtain very wrong results.

**Questions:**

- Can you please explain why BestMatch (Equation 1) does not consider the instance $x$ being evaluated when computing the similarity between the labels?

---

> ### Author Response · Authors · 2025-11-20
> **Point-by-point rebuttal**
>
> Dear reviewer, we sincerely thank you for taking the time to review our 30+ page paper, we recognize the time and effort it must have took. Please find a point-by-point response to your comments below:
>
> - Formulation: We've clarified the problem formulation in Section 3.1, now explicitly showing (x, y) sampled from P(X, Y). The set-based notation is standard for multi-label problems where predictions have arbitrary size and no ordering.
> - Context-dependent similarity: You raise an interesting point. While per-instance adaptation would be circular (we'd need to know the correct classification to adjust similarities to then measure the quality of the classification), per-cohort adaptation is a promising direction. For example, adjusting similarities based on annotator subgroups or domain subcategories. We now discuss this in Section C as future work.
> - Synthetic experiments: The synthetic settings give us precise control over classifier error, allowing provable claims about metric behavior. We agree connecting to real classification would strengthen this. Our real-data experiments (Sections 4.3.3 onwards, as well as Appendix results) provide that grounding, showing consistent results across 8 datasets.
> - Code: Semantic F1 (https://anonymous.4open.science/r/semantic-f1-score-B4C3/), and separately a repo for its validation (https://anonymous.4open.science/r/semantic-f1-score-validation-63A6/) have been anonymized.
> - Overselling: Fair point. We've softened claims in the revised manuscript, making it conditional to appropriate similarities being used. Our results support that Semantic F1 is more informative in domains with fuzzy boundaries, but we now frame this more carefully. Let us know if more statements need care.
>
> --Edited for clarity --

---

> > ### Comment · Reviewer_hnNu · 2025-11-27
> >
> > I would like to thank the authors for their comments and clarifications. I have no further questions to ask. After reading the other reviews, I have updated my evaluation (taking into consideration my limited confidence on the topic).

---

### Official Review · Reviewer_TjbF · 2025-11-08

**Soundness:** 2
**Presentation:** 3
**Contribution:** 2
**Rating:** 4
**Confidence:** 3

**Summary:**

This paper introduces Semantic F1 Scores, a family of evaluation metrics for multi-label classification that account for semantic similarity between labels. The core idea is to give partial credit when a predicted label is semantically related to the true label, instead of counting it as a complete miss as in standard F1. The metric uses a two-step matching (predict-to-true and true-to-predict) to compute “semantic precision” and “semantic recall,” which are combined into an F1 score. Semantic F1 is designed to be backward-compatible (it reduces to standard F1 when no partial credit is allowed). Additionally, to theoretical arguments, the authors conduct an empirical study on eight datasets (synthetic and real) and demonstrate that Semantic F1 provides potentially fairer and more informative evaluation in tasks with subjective or overlapping labels, correlating more closely with real-world outcomes than classical F1.

**Strengths:**

- The paper is easy to read and understand.
- Proposes a semantic F1 measure that is nicely motivated and theoretically elegant.
- The experimental study kind of demonstrates some desirable properties of semantic F1 over prior single-step and Hungarian-style approaches.

**Weaknesses:**

- I think the biggest weakness of the approach is dependency on the similarity matrix. If the matrix S is poorly specified or biased, scores could be misleading or unfair, which makes it tricky to use as an evaluation metric. The defining similarity may be non-trivial. Similarity derived from label co-occurrence or embeddings might not truly reflect conceptual closeness. Results with different S matrices are not comparable.
- Baseline comparison on real tasks is limited, as on real datasets, evaluation focuses on hard F1 vs Semantic F1, with no direct comparison to alternative semantic metrics, which also feels limited in a synthetic study.
- I'm generally a bit confused by experiments; it is not clear to me what is optimized and what is evaluated, and why should I care that Semantic F1 better correlates with the task than hard F1? And what about other soft/semantic metrics?
- Many labels on Figures 2 and 3 are so small that they are not readable when printed.

**Questions:**

Please, see weaknesses session.

---

> ### Author Response · Authors · 2025-11-20
> **Point-by-point rebuttal**
>
> Dear reviewer, we want to again thank and recognize the time and effort you have committed to this work, especially in our case with our extensive 30+ pages. Please find a point-by-point response to the points you raised below.
>
> - Similarity matrix dependency: As discussed in our general comment, we alleviate this significantly, our method maintains discriminative power even with 60% noise (Figures 2b, 15).
> - Different matrices not comparable: You're right that absolute values differ across matrices. However, relative model rankings remain stable across reasonable matrix choices. As stated above, Figures 2b, 15 demonstrate so. We've added discussion of this in the Conclusion, advocating for uniformity of similarity matrices across different works. We will provide extra quantification with additional experiments soon.
> - Baselines: We've added comparisons to other semantic-based metrics in Section F.6. Results show that single-step approaches (precision-only or recall-only) are unreliable. For example, semantic precision alone has a correlation of down to 0.15, while Semantic F1 provides consistent evaluation across all tested domains.
> - Experiment clarity: We optimized models using standard training losses, as described in the Implementation sections, or used pretrained LLMs, then evaluated using different metrics. The question is: which evaluation metric better reflects real-world usefulness? Our experiments show Semantic F1, e.g., correlates more strongly with downstream outcomes than hard F1. We have improved the commentary throughout Section 4.3 (we include revised and explicit “Motivation” paragraphs) and the appendix results to better explain the usefulness of each experiment.
> - Figure readability: Fixed, thank you. We will continue to try to improve the readability of the figures while respecting the page limit.
>
> --Edited for clarity--

---

> > ### Author Response · Authors · 2025-11-22
> > **Updated results**
> >
> > Dear reviewer, we have completed the additional promised experiments regarding your comment about different matrices not being comparable. We found that across noise levels in the similarity matrix, the ranking of classifiers by Semantic F1 remains the same. This means that, while the value of the metric is not the same as you correctly pointed out, it maintains its informativeness with somewhat well-specified, but far from perfect, similarities. For example, the rankings between ideal S and $S_a$ with $a=0.4$ (60% noise) have a correlation of 0.989, averaged across all examined settings. We have added these results in Section F.1 (Figure 12).

---

### Author Response · Authors · 2025-11-20
**General comment to all reviewers**

Thank you for taking the time to thoughtfully engage with our work! We want to recognize the contributions and the effort all reviewers put towards the review of our paper. Several reviewers raised concerns about similarity matrix dependency, which we wanted to address in a general comment:

While we don't fully solve this challenge, we alleviate it significantly compared to prior work: Figures 2b and 15 demonstrate that our method maintains discriminative power even with 60% noise in the similarity matrix.
Moreover, the domains we examine have well-established similarity structures from decades of research (e.g., Plutchik's wheel for emotions, established moral foundations theory for morality, for example establishing loyalty and authority as more similar than loyalty and fairness). We are not asking practitioners to construct matrices from scratch.
Finally, this dependency is inherent to all semantic evaluation, including the cited published work. We unify existing approaches (hierarchies, embeddings, graphs) under a common framework rather than introducing new assumptions.
We have also conducted additional experiments as requested. Section F.6 now includes comparisons with other semantic-based baselines on real data, showing that precision or recall alone can be unreliable, while Semantic F1 provides consistent evaluation across settings. In particular, the correlation between objective and subjective settings can be as low as 0.15 when measured with semantic precision alone, significantly lower than Semantic F1.

We would greatly appreciate it if these points are taken into consideration in revising your opinion on our paper during this rebuttal period.

--Edited for clarity--

---

### Meta-Review · Area_Chair_zV9a · 2026-01-06

**Summary:**

The authors introduce Semantic F1 Scores as evaluation metrics for multi-label classification, where classes are semantically related and boundaries between them are "fuzzy". Unlike conventional F1, which penalises semantically related predictions as fully incorrect, Semantic F1 uses a label similarity matrix to compute soft precision- and recall-like measures. Its two-step formulation allows comparison of label sets of arbitrary size without forced matching, awarding partial credit for related labels. This is claimed to yield fairer and more realistic evaluations in domains with overlapping categories or annotator disagreement.

The reviews on the paper are mixed. Although the reviewers recognise the relevance of the topic and consider the authors' approach to be interesting and potentially useful, they nevertheless express a number of criticisms regarding the measure's dependency on the similarity matrix, comprehensiveness and conclusiveness of experimental studies, related work and lack of novelty, among others. The authors could clarify some of the issues raised in the reviews, and one reviewer was also willing to adapt their score. Overall, however, the reviewers are still leaning toward rejection. In conclusion, the paper does not meet the standards and high requirements of the conference in its current form and therefore cannot be recommended for acceptance.

**Reviewer Concerns:**

I think the more technical comments have been addressed, but general criticism such as lack of novelty more more conceptual issues such as dependency on similarity relation are of course more difficult to deal with.

**Reviewer Scores:**

One review (with original rating 2) raised their score, another one (rating 4) kept it. For the others, it's difficult to say, but I would consider a significant increase of the scores as rather unlikely.

---

### Decision · Program_Chairs · 2026-01-26

Reject